



# Mg/Ca and $\delta^{18}$O in living planktic foraminifers from the Caribbean, Gulf of Mexico and Florida Straits

Anna Jentzen[1,a], Dirk Nürnberg[1], Ed C. Hathorne[1], Joachim Schönfeld[1]

[1]GEOMAR Helmholtz Centre for Ocean Research Kiel, Wischhofstrasse 1–3, 24148 Kiel, Germany
[a]now at: Max Planck Institute for Chemistry, Hahn-Meitner-Weg 1, 55128 Mainz, Germany

*Correspondence to*: Anna Jentzen (anna.jentzen@mpic.de)

**Abstract.** Past ocean temperatures and salinities are successfully approximated from combined stable oxygen isotopes ($\delta^{18}$O) and Mg/Ca measurements in fossil foraminiferal tests. To further refine this approach, we collected living planktic
foraminifers by net sampling and pumping of seasurface waters from the Caribbean Sea, the eastern Gulf of Mexico, and Florida Straits. Analyses of $\delta^{18}$O and Mg/Ca in eight living planktic species (*Globigerinoides sacculifer*, *Orbulina universa*, *Neogloboquadrina dutertrei*, *Pulleniatina obliquiloculata*, *Globorotalia menardii*, *Globorotalia ungulata*, *Globorotalia truncatulinoides* and *Globorotalia tumida*) were compared to measured in situ properties of the ambient seawater (temperature, salinity and $\delta^{18}$O$_{seawater}$) and fossil tests of underlying surface sediments. "Vital effects" such as symbiont
activity and test growth cause $\delta^{18}$O disequilibria to the ambient seawater and a large scatter in foraminiferal Mg/Ca. Overall, ocean temperature is the most prominent environmental influence on $\delta^{18}$O$_{calcite}$ and Mg/Ca. Enrichment of the heavier $^{18}$O isotope in living specimens below the mixed layer and in fossil tests are clearly related to lowered in situ temperatures and gametogenic calcification. Mg/Ca-based temperature estimates of *G. sacculifer* indicate seasonal maximum accumulation rates on the seafloor in early spring (March) at Caribbean stations and later in the year (May) in the Florida Straits, related to
the respective mixed layer temperatures of ~26 °C. Notably, *G. sacculifer* reveals a positive linear relationship between foraminiferal derived $\delta^{18}$O$_{seawater}$ estimates and both measured in situ $\delta^{18}$O$_{seawater}$ and salinity. Our results affirm the applicability of existing $\delta^{18}$O and Mg/Ca calibrations for the reconstruction of past ocean temperatures and $\delta^{18}$O$_{seawater}$ reflecting salinity due to the convincing accordance of proxy data in both living and fossil foraminifers, and in situ environmental parameters. Large "vital effects" and seasonally varying proxy signals, however, need to be taken into
account.

## 1 Introduction

Calcite tests of planktic foraminifers are precipitated from the surrounding seawater and their stable oxygen isotope compositions ($\delta^{18}$O$_{calcite}$) and Mg/Ca ratios are established proxies to reconstruct past ocean conditions (e.g. Erez and Luz, 1983; Nürnberg et al., 2000). The $\delta^{18}$O$_{calcite}$ signature depends on the ambient seawater temperatures and oxygen isotopic
compositions ($\delta^{18}$O$_{seawater}$) the planktic organism is thriving in. Their relationship was defined in several $\delta^{18}$O-paleotemperature equations (e.g. Erez and Luz, 1983; Bouvier-Soumagnac and Duplessy, 1985; Bemis et al., 1998). Earlier studies showed that $\delta^{18}$O$_{calcite}$ reveals an offset to the equilibrium of the seawater, caused by environmental factors (e.g. salinity, carbonate ion concentration [CO$_3^{2-}$], ocean pH) and/or biological controlled processes, so-called "vital-effects" (Weiner and Dove, 2003) (e.g. symbionts photosynthesis, respiration) as influencing factors (Spero and Lea, 1993; Spero et
al., 1997; Bemis et al., 1998; Bijma et al., 1999).
Mg/Ca ratios in foraminiferal tests are predominantly controlled by ocean temperature. Meanwhile, robust planktic species-specific calibrations exist (e.g. Nürnberg, 1995; Nürnberg et al., 1996; Lea et al., 1999; Anand et al., 2003; Regenberg et al., 2009), which allow to reconstruct the thermal structure of the entire water column, even on timescales of millions of years.



The incorporation of magnesium during calcification is largely driven by physiological processes, which may cause Mg/Ca heterogeneity in single tests with high and low Mg-bands in some species (Erez, 2003; Sadekov et al., 2005; Bentov and Erez, 2006; Hathorne et al., 2009; Spero et al., 2015). Further, environmental parameters (e.g. salinity, $[CO_3^{2-}]$, ocean pH) may affect foraminiferal Mg/Ca (Nürnberg et al., 1996; Lea et al., 1999; Russel et al., 2004; Kisakürek et al., 2008). Most

critical are carbonate dissolution processes that considerably lower Mg/Ca in foraminiferal tests (Brown and Elderfield, 1996; Rosenthal et al., 2000; Regenberg et al., 2006).

Relatively few (isotope) geochemical studies were conducted on recent/living planktic foraminifers, either collected from the water column or cultured under controlled laboratory conditions. These studies are an important addition to a multitude of core-top and downcore studies, allowing us to assess the different controlling factors on $\delta^{18}O_{calcite}$ and Mg/Ca during

biomineralization (e.g. Kahn, 1979; Erez and Honjo, 1981; Nürnberg et al., 1996; Lea et al., 1999; Russel et al., 2004; Kisakürek et al., 2008; Spero et al., 2015).

We here systematically sampled the upper water column of the Caribbean, the eastern Gulf of Mexico, and Florida Straits for living tropical and subtropical planktic foraminifers using plankton nets and on board pumping devices. $\delta^{18}O_{calcite}$ and Mg/Ca analyses within bulk calcite and single chambers of living specimens collected from different depth intervals were i) related

to ocean parameters (temperature, salinity, $\delta^{18}O_{seawater}$) measured in water samples from CTD sampling stations nearby, and ii) compared to fossil counterparts from underlying or nearby surface sediments. Our integrated approach aims to evaluate (i) "vital-effects" under natural conditions, (ii) the ontogenetic development in particular test growth and (iii) the impact of environmental conditions on foraminiferal $\delta^{18}O_{calcite}$ and Mg/Ca to further substantiate their potential as paleoceanographic proxies.

## 2 Material and Methods

### 2.1 Sampling and preparation of planktic foraminifers

Analyses were performed on living and fossil foraminifers sampled from plankton nets, pumping from below the ship, and surface sediments obtained during cruises SO164 (RV *Sonne*) in May/June 2002 (Nürnberg et al., 2003) and M78/1 (RV *Meteor*) in February/March 2009 (Schönfeld et al., 2011) (Fig. 1; Table 1). To collect living planktic foraminifers, the

Hydrobios Midi multiple opening-closing plankton net (MSN) with a mesh size of 100 μm was deployed at five stations in different water depth intervals (surface to max. 400 m) (Table 1). Further sampling of living specimens was accomplished by pumping seawater from 3.5 m water depth during ship´s transit and subsequent filtering over a 63 μm sieve (PF samples). Immediately after sampling, the plankton samples (MSN and PF) were preserved in a mix of 50 % ethanol and seawater. The MSN samples were stained with Rose Bengal (2 g/l). Surface sediment samples were recovered by Multicorer and USNEL

giant box corer at positions close to the MSN stations (Table 1). During cruise M78/1, salinity and temperature were recorded by the RBR XR-420 Conductivity-Temperature-Depth (CTD) profiler and by the shipboard thermosalinograph (Fig. 2). For stable isotope analyses in seawater ($\delta^{18}O_{seawater}$), water samples were collected at different water depths (Table 1) with the shipboard rosette Niskin bottle system connected to the CTD profiler, filled in glass bottles (100 ml) and poisoned with 0.2 ml $HgCl_2$ to prevent biological activity.

In the laboratory (GEOMAR, Kiel), the plankton net samples were rinsed with tap water and all foraminifers were picked wet with a glass pipette. The picked foraminifers were dried on a filter paper at room temperature, fractionated into different mesh sizes (100–125, 125–150, 150–250, 250–300, 300–400, 400–500 and >500 μm) and identified on species level after Bé (1967) and Schiebel and Hemleben (2017). For isotope and geochemical analyses, individual tests from eight different species were selected including: *Globigerinoides sacculifer* (i.e., *Trilobatus sacculifer*; Spezzaferri et al., 2015) with a

spherical last chamber, *Orbulina universa, Neogloboquadrina dutertrei, Pulleniatina obliquiloculata, Globorotalia menardii, Globorotalia ungulata, Globorotalia truncatulinoides* dextral, and *Globorotalia tumida* (Supplement S5). Only



cytoplasm-bearing specimens with an intact calcite test were considered for analyses, indicating that the foraminifers were still alive when collected. For all species, the weighted average living depth (m) and habitat (=living) temperature (°C) (temperature at the weighted average living depth) was calculated based on standing stocks (individual m$^{-3}$) in the water column (Table 2; cf. Jentzen et al., submitted).

Surface sediment samples were freeze-dried, wet sieved using tap water over a 63 µm sieve, and dried at 40 °C. Single intact tests were picked from the 355–400 µm size fraction, to be directly comparable with published data from similar Caribbean station sites (existing $\delta^{18}O_{calcite}$ data from Steph et al., 2009 and Mg/Ca data from Regenberg et al., 2006).

**2.2 Stable isotope analyses**

Depending on the selected species and size fraction, a varying number of specimens were analysed for stable isotopes
($\delta^{18}O_{calcite}$ and $\delta^{13}C_{calcite}$) (cf. Supplement S1). Prior to the measurements, the foraminiferal tests were cracked and the remaining cytoplasm was removed with a needle. The measurements were run on a ThermoScientific MAT 253 mass spectrometer connected to an automatic carbonate preparation device Kiel CARBO IV at GEOMAR. The stable isotope results are reported relative to the Vienna Pee Dee Belemnite (VPDB) in per mil (‰) and calibrated versus the National Bureau of Standards (NBS) 19. The in house standard (Solnhofen limestone) run multiple times with every magazine of
samples and gives a long-term analytic precision of <0.06 ‰ (±1σ) for $\delta^{18}O_{calcite}$ and <0.03 ‰ (±1σ) for $\delta^{13}C_{calcite}$, respectively.

Stable oxygen isotopes in seawater ($\delta^{18}O_{seawater}$) were analysed by the Isotope Ratio Infrared Spectroscopy (IRIS) analyser (Model L1102-i CRDS) at the laboratory of GeoZentrum Nordbayern (Erlangen) (Van Geldern and Barth, 2012). The measurements are expressed in per mil (‰) versus the Vienna Standard Mean Ocean Water (VSMOW). The analytical
precision is better than 0.05 ‰ (±1σ).

The difference between the predicted inorganic calcite $\delta^{18}O$ signal of the seawater (calcite formed in thermodynamic equilibrium, $\delta^{18}O_{equilibrium}$) and the $\delta^{18}O_{calcite}$ value of the foraminifer is commonly termed the "vital effect" ($\delta^{18}O_{disequilibrium}$) (Table 2):

$$\delta^{18}O_{disequilibrium} = \delta^{18}O_{calcite} - \delta^{18}O_{equilibrium} \tag{1}$$

To determine $\delta^{18}O_{equilibrium}$ (Fig. 3a), the temperature equation of Kim and O´Neil (1997) for inorganic precipitation was applied follow the relationship:

$$\delta^{18}O_{equilibrium} = 25.778 - 3.333 * (43.704 + T)^{0.5} + \delta^{18}O_{seawater} \tag{2}$$

with in situ temperatures (°C) measured during cruise M78/1 by CTD and measured seawater ($\delta^{18}O_{seawater}$) values (Schönfeld et al., 2011; Supplement S1). $\delta^{18}O_{seawater}$ was corrected to the PDB scale by subtracting 0.27 ‰ after Hut (1987).

**2.3 Mg/Ca analyses**

Mg/Ca ratios in foraminiferal calcite were analysed from both bulk samples comprising numerous of tests of a single species, and single specimens, depending on their abundances (cf. Supplement S1). Prior to analyses, the samples were cleaned with a hydrogen peroxide-cleaning step following Barker et al. (2003), which is suggested to be an efficient method to remove the high amount of cytoplasm in live foraminifers (Pak et al., 2004). We omitted a reductive hydrazine cleaning
step as this step is unnecessary for plankton samples. Furthermore, employing only the oxidative cleaning step allows for direct comparison to foraminiferal Mg/Ca from surface sediments, which are treated similarly (Regenberg et al., 2006). For each bulk sample (plankton net and sediment), ~400–800 µg of *G. sacculifer*, *N. dutertrei* and *G. ungulata* from different size fractions were used for analyses (Supplement S1). The tests were gently crushed between two glass plates, in order to open the chambers, and transferred into a vial. The samples were first rinsed with ultrapure water and ethanol, including an





ultrasonic treatment. Then, 250 µl of a NaOH/H$_2$O$_2$ solution (100 µl 30 % H$_2$O$_2$ and 10 ml NaOH) were added to each vial and placed for 20 minutes in a hot water bath (92 °C). For the plankton samples these steps were repeated 1–2 times in order to completely remove the cytoplasm. The samples were subsequently rinsed with ultrapure water. Finally, the tests were leached with 250 µl of HNO$_3$ (0.001 M). Prior to the element analyses, the samples were dissolved in HNO$_3$ (0.075 M). The

measurements were performed with an axial-viewing VARIAN 720 Inductively Coupled Plasma-Optical Emission Spectrometer (ICP-OES) at GEOMAR. The data of the measurements were normalised and trend-corrected using the ECRM 752-1 standard (3.761 mmol mol$^{-1}$ Mg/Ca; Greaves et al., 2008). The analytic precision is 0.1 mmol mol$^{-1}$ (±2σ).

Single chambers of live collected foraminifers were analysed with an Excimer ArF 193 nm laser ablation system, coupled to an Inductively Coupled Plasma-Mass Spectrometer (ICP-MS Agilent 7500cx) at GEOMAR. Single foraminifers were

cleaned with a buffered hydrogen peroxide solution, in a similar way as the bulk samples. Only one specimen was put into a vial to avoid breaking the test during the cleaning process. Each test was rinsed with ultrapure water and ethanol before adding 250 µl of NaOH/H$_2$O$_2$ solution. The samples were then placed in a hot water bath (92 °C) for 20 minutes and rinsed with ultrapure water and ethanol afterwards. Subsequently, the samples were dried at room temperature. The laser ablation technique allowed us to ablate through the test wall from the outer test surface towards the inner side. Its spot size diameter

was focused to 50 and 75 µm. Ablation profiles were carried out on the last four chambers (F to F-3) (Supplement S1). The energy density of the laser was 0.9–2.6 J cm$^{-2}$ and a laser repetition rate of 5 and 7 Hz was selected. The following isotopes were measured: $^{24}$Mg, $^{26}$Mg, $^{27}$Al, $^{43}$Ca, $^{44}$Ca, $^{55}$Mn, $^{66}$Zn, $^{88}$Sr, $^{232}$Th and $^{238}$U. The ablation was stopped when the test wall was penetrated. Analyses were calibrated using standard glasses 610 and 612 of National Institute of Standards and Technology (NIST) using the values of Jochum et al. (2011). The NIST 610 and NIST 612 were ablated with an energy

density of 2–3 J cm$^{-2}$ after every ten measurements of foraminiferal tests. Raw counts of elements were processed offline and $^{43}$Ca was used as internal standard to account for ablation yield. Outliers (average value ±2σ) were rejected from the results. A powder pellet of JCt-1 (giant clam shell) was used as reference and repeatedly analysed (n=15) during the ablation sessions revealing an average Mg/Ca ratio of 1.21 ± 0.13 mmol mol$^{-1}$ (standard deviation of 10.6 %, 1σ) being consistent with the consensus of solution analyses in many laboratories (Mg/Ca=1.289 mmol mol$^{-1}$ Hathorne et al., 2013).

In situ temperatures (°C) measured during cruise M78/1 (Schönfeld et al., 2011) were compared to derived Mg/Ca-temperature estimates. We applied different calibrations for each species to account for species-specific differences (e.g. Russel et al., 2004; Cléroux et al., 2008; Regenberg et al., 2009; cf. Supplement S2).

### 2.4 Calculation of δ$^{18}$O$_{seawater}$

The combination of δ$^{18}$O$_{calcite}$ and Mg/Ca in foraminiferal tests allows us to estimate δ$^{18}$O of the ambient seawater (Craig and

Gordon, 1965; Schmidt, 1999; Fig. 3b), which is used as a proxy for surface seawater salinity. We compared our measured in situ δ$^{18}$O$_{seawater}$ to δ$^{18}$O$_{seawater}$ estimates derived from combined foraminiferal δ$^{18}$O$_{calcite}$ and Mg/Ca-temperatures of *G. sacculifer*. For the calculation we used the species-specific δ$^{18}$O-paleotemperature equation for *G. sacculifer* of Spero et al. (2003) with the species-specific Mg/Ca-temperature calibration for *G. sacculifer* of Regenberg et al. (2009).

### 2.5 Calcite dissolution

Calcite dissolution can affect foraminiferal Mg/Ca as a function of the regionally different calcite saturation states in the oceans and the sensitivity of the species-specific test structure (Brown and Elderfield, 1996; Regenberg et al., 2006; 2014). The calcite saturation state Δ[CO$_3^{2-}$] is defined as:

$$\Delta[CO_3^{2-}] = [CO_3^{2-}]_{in-situ} - [CO_3^{2-}]_{saturation} \qquad (3)$$

and decreases from the surface (~150–200 µmol kg$^{-1}$) to ~5000 m water depth (<0 µmol kg$^{-1}$) in the eastern Caribbean Sea

and Gulf of Mexico (Fig. 4). Δ[CO$_3^{2-}$] of ~21 µmol kg$^{-1}$, which is a critical threshold for the onset of selective Mg$^{2+}$ ion





removal from planktic foraminiferal calcite, is at ~2500–3000 m water depth in the study area. Below this, the undersaturated waters generally lower foraminiferal Mg/Ca through preferential dissolution (Regenberg et al., 2006; 2014). As all plankton net samples of this study were taken from shallower than 400 m water depth, the studied living foraminifers originate from supersaturated seawater with respect to calcite ($\Delta[CO_3^{2-}] > 50\ \mu mol\ kg^{-1}$) and that substantial $Mg^{2+}$ ion

removal (loss of higher Mg/Ca calcite) is not to be expected. For fossil tests from surface sediments below 2500–3000 m water depth we use the dissolution corrected Mg/Ca values from Regenberg et al. (2006; 2009) (cf. Supplement 1).

### 3 Results and Discussion

#### 3.1 Hydrographical setting during sampling

In order to be able to directly relate our results on vertical foraminiferal distribution patterns and species-specific (isotope)

geochemical signatures to the modern hydrographic conditions in the study area, we also took temperature, salinity and $\delta^{18}O_{seawater}$ measurements. The CTD and thermosalinograph data gathered during cruise M78/1 (February–March 2009) reveal low sea surface temperatures (SST) in the Gulf of Mexico (~20 °C) and Florida Straits (~24 °C) (Fig. 1; 2) comparable to the boreal winter situation (Fig. 2; Locarnini et al., 2013). Hydrographic conditions in the Caribbean vary seasonally with a large range of SSTs (range in the Florida Straits up to 5 °C) and salinities (SSS; range in the Caribbean Sea

up to 1 (psu)) (Fig. 2) and are closely linked to the migrating Intertropical Convergence Zone (ITCZ), which is at its northernmost position (6–10 °N) during summer (Locarnini et al., 2013; Zweng et al., 2013). The surface mixed layer extends to max. 100 m water depth in the Caribbean and is characterised by the relatively fresh Caribbean Water (CW; <36 psu). The lowest salinity is recorded in the southeastern Caribbean during summer and autumn when the Amazon and Orinoco river discharge is most intense and freshwater plumes arrive in the Caribbean Sea (Wüst, 1964; Müller-Karger et al.,

1989; Chérubin and Richardson, 2007). Modified CW is transported via anticyclonic eddies (Loop Current) towards the Gulf of Mexico and Florida Straits (Vukovich, 2007). In the upper thermocline, the highly saline Subtropical Under Water (SUW; >37 (psu)) prevails. This water mass originates in tropical and subtropical regions (Gallegos, 1996; Blanke et al., 2002) and resides in ~80–160 m water depth. The 18 °C Sargasso Sea Water (Eighteen Degree Water = EDW) prevails in ~200–400 m water depth entering the Caribbean Sea via the passages of the Greater Antilles (Morrison and Nowlin, 1982). The Gulf

Common Water (~23 °C and ~36.4 (psu); Vidal et al., 1994) possibly influences the Florida Straits hydrography (Station 210/211) in the upper thermocline at 100–150 m, characterised by low salinity (36.5 (psu)).
Seawater $\delta^{18}O$ ($\delta^{18}O_{seawater}$) averages to ~0.9 ‰ (VSMOW) in the uppermost 400 m water depth (Fig. 3a). Highest $\delta^{18}O_{seawater}$ values (1.3 ‰) can be found in the salinity maximum at ~60–150 m water depth, whereas the lowest value (0.3 ‰) is measured in the deepest sample at the lowest salinity. Additionally, the in situ $\delta^{18}O_{seawater}$ and salinity recorded during M78/1

show a positive correlation (linear regression, r = 0.81) and yield similar values as earlier data sets from the Caribbean Sea (Schmidt et al., 1999) (Fig. 3b). The $\delta^{18}O_{equilibrium}$ increases with depth from ~-1.5 to 1 ‰ in dependence of the decreasing ocean temperature (Fig. 2; 3a).

#### 3.2 Vital effects on foraminiferal $\delta^{18}O_{calcite}$

In order to address the effects of symbiont activity and life cycle on the foraminiferal oxygen isotopes, $\delta^{18}O_{calcite}$ values of

living foraminifers were compared to the calculated $\delta^{18}O_{equilibrium}$ of the ambient seawater and $\delta^{18}O_{calcite}$ estimates of fossil tests from underlying surface sediments.





### 3.2.1 Symbionts and life cycle effect on foraminiferal $\delta^{18}O$ and $\delta^{13}C$

Specimens of *G. sacculifer* and *O. universa* from the mixed layer are characterised by large negative $\delta^{18}O_{disequilibrium}$ values of -0.35 ‰ and -0.32 ‰, respectively (Table 2). These two species host dinoflagellates as symbionts (Gastrich, 1987) and similarly negative $\delta^{18}O_{disequilibrium}$ values were reported in spinose, symbiont-bearing species caught in plankton tows from
various ocean areas (Table 2 and references therein). Laboratory experiments (Spero, 1992; Spero and Lea, 1993; Bemis et al., 1998) revealed a depletion of 0.3 to 0.6 ‰ in $\delta^{18}O_{calcite}$ of *O. universa* and *G. sacculifer* under high irradiance levels related to algae photosymbiont activity. In particular, a high irradiance in the euphotic zone intensifies the photosynthetic rate in the Caribbean Sea under its prevailing oligotrophic conditions (Spero and Parker, 1985; Morel et al., 2010). Enhanced photosymbiont activity increases the $O_2$ concentration and fosters $CO_2$ fixation, resulting in an elevated pH within the
microenvironment around the living foraminifer (Jørgensen et al., 1985; Rink et al., 1998). Both, increasing pH and increasing carbonate ion concentration $[CO_3^{2-}]$ apparently cause a depletion of $\delta^{18}O_{calcite}$ (Spero et al., 1997; Bijma et al., 1999).

Among all species studied, only *G. sacculifer* and *N. dutertrei* reveal a significant positive correlation (Spearman rank correlation, $p<0.05$) between test size and stable isotopes ($\delta^{18}O_{calcite}$ and $\delta^{13}C_{calcite}$) (Fig. 5, Supplement S3), suggesting that
ontogeny affects the isotopic fractionation processes. The species *G. ungulata* shows lower $\delta^{18}O_{calcite}$ values in the test size fraction <300 µm and *G. menardii* indicate no significant ontogenetic effect ($p>0.5$; Fig. 5). It should be noted that for some species we did not have enough sample material in all test size classes. However, our results are consistent to Kahn (1979), Kahn and Williams (1981), Spero and Lea (1996) and Bemis et al. (1998), who postulated that juvenile foraminifers have a larger "vital-effect" than adult individuals, with their tests being depleted of the heavy [18]O and [13]C isotopes due to a higher
metabolic rate (incorporation of respired $CO_2$) and/or rapid growth rate. Rapidly growing calcitic skeletons result in a stronger kinetic isotope fractionation and cause the depletion of heavier [18]O and [13]C isotopes (McConnaughey, 1989).

Vertical migration of planktic species to deeper and colder water masses during their life cycle may additionally affect $\delta^{18}O_{calcite}$, leading to commonly higher values in adult specimens (Kroon and Darling, 1995; Lončarić et al., 2006; Birch et al., 2013). Samples from the same test size fraction of all species exhibit the enrichment of heavier [18]O isotopes at deeper
water levels (Fig. 6; Table 3). We speculate that the increasing $\delta^{18}O_{calcite}$ at deeper water levels is a function of increasing $\delta^{18}O_{equilibrium}$ of the ambient seawater, rather than ontogenetic effects itself. The surface dweller *G. sacculifer* reveals the largest $\delta^{18}O_{disequilibrium}$ value (~1 ‰) in the thermocline (Table 2). As a higher rate of photosynthetic processes in deeper water depths can be excluded and specimens were still alive when sampled, we suggest that *G. sacculifer* completed calcifying in the thermocline before reproduction. Our observation corroborates South Atlantic plankton net studies of
Lončarić et al. (2006), who noted that *G. sacculifer* $\delta^{18}O_{calcite}$ increased with depth in the upper 60 m water depth and remained constant below the surface mixed layer, even though $\delta^{18}O_{equilibrium}$ increased continuously.

### 3.2.2 The $\delta^{18}O$ offset between living and fossil foraminifers

It becomes evident that almost all fossil tests from surface sediment samples, in particular *N. dutertrei*, *P. obliquiloculata*, *G. truncatulinoides* and *G. tumida* are enriched in $\delta^{18}O_{calcite}$ (>0.5 ‰) compared to their living counterparts from the water
column (Fig. 6; Table 3). $\delta^{18}O_{calcite}$ of fossil shallow dwellers *G. sacculifer* and *O. universa* are rather close to those values of specimens caught in the thermocline (average difference of 0.14 ‰ and 0.02 ‰, respectively) (Table 3). Yet, the overall discrepancy in $\delta^{18}O_{calcite}$ between fossil and living specimens may be best explained by gametogenetic calcification processes or calcite crust formation, which take place during the vertical migration through the water column. At the end of the life cycle and prior to gametogenesis, various planktic foraminifer species (including *G. sacculifer*, *O. universa*,
*P. obliquiloculata*, *G. truncatulinoides*, *G. tumida*) add a calcitic crust of variable thickness on the outer surface of the test (Schiebel and Hemleben 2017, and references therein). Based on calculations of Bouvier-Soumagnac and Duplessy (1985)





and Hamilton et al. (2008) up to 25 % (~4 μg) gametogenic calcite is added by *O. universa*, which is mainly secreted in colder waters prior to reproduction. The tests thereby lose their glassy and transparent appearances (Bé, 1980; Deuser et al., 1981; Duplessy et al., 1981b; Hemleben et al., 1985; Schweitzer and Lohmann, 1991). Specifically, spinose species resorb their spines before releasing their gametes (Bé and Anderson, 1976; Spero, 1988). These processes result in heavier $\delta^{18}O_{calcite}$

compositions of fossil tests from surface sediments (and even individual foraminifers from sediment traps) (Duplessy et al., 1981b; Bouvier-Soumagnac and Duplessy, 1985; Bouvier-Soumagnac et al., 1986; Lin et al., 2011). Consistently, the heavy $\delta^{18}O_{calcite}$ values in adult specimens of *G. truncatulinoides* and *G. tumida* may be best explained by vertical migration into colder water masses at a late ontogenetic stage (Franco-Fraguas et al., 2011; Birch et al., 2013). Orr (1967) and Vergnaud-Grazzini (1976) recognised that living individuals of *G. truncatulinoides* with a thick test and pustules on the test surface are

more likely to be found in deeper water masses than non-ornamented, thin-shelled specimens. As expected, such tests had $\delta^{18}O_{calcite}$ values close to those observed in surface sediments. Overall, our proxy database supports the notion that specimens of *P. obliquiloculata*, *G. tumida* and *G. truncatulinoides* add a thick opaque calcite layer or cortex at deeper water depths than ~400 m. Hence, the fossil tests are enriched in $\delta^{18}O_{calcite}$ relative to the living foraminifers (up to 0.85 ‰) (Fig. 6; Table 3).

During the sampling campaign in February/March 2009, mainly juvenile specimens of *N. dutertrei* were found in plankton nets (mode test size fraction 150–250 μm; Jentzen et al., submitted). This finding may additionally explain the large $\delta^{18}O_{calcite}$ offset between living foraminifers and fossil tests (~1 ‰) (Fig. 6; Table 3). Kroon and Darling (1995) recognised that small specimens of *N. dutertrei* have similar $\delta^{18}O_{calcite}$ values as surface dwellers and lower values than large specimens, supporting the notion on the ontogenetic related migration to deeper waters. Fairbanks et al. (1982) and Bouvier-Soumagnac

and Duplessy (1985) also noted increasing $\delta^{18}O_{calcite}$ values of *N. dutertrei* with increasing water depth in the Panama Basin and Indian Ocean, suggesting that this species secrete substantial proportions of their tests below the mixed layer. Furthermore, living *N. dutertrei* from the South China Sea were depleted in $\delta^{18}O_{calcite}$ compared to individuals from sediment traps (Lin et al., 2011). Our data confirm these assumptions as we recognised higher $\delta^{18}O_{calcite}$ values and larger individuals of *N. dutertrei* in surface sediments compared to the mixed layer (Fig. 6; Table 3; Jentzen et al., submitted).

The species *G. menardii* show increasing $\delta^{18}O_{calcite}$ values from the mixed layer to the thermocline (+0.3 ‰), and from the thermocline to the surface sediments (+0.2 ‰) pointing to decreasing ambient seawater temperatures at deeper water levels and migration within the water column (Fig. 2; Table 3). Apparently, *G. ungulata* is an exception to the rule, as this species does not show the enrichment of $\delta^{18}O_{calcite}$ in fossil tests compared to living specimens (Fig. 6; Table 3). Yet, the species secreted their calcite tests close to the equilibrium with the ambient seawater (0.01–0.08 ‰) throughout the water column

(Table 2). The average surface sediment $\delta^{18}O_{calcite}$ value corresponds well with the depth where the highest standing stock was observed during the sampling campaign in February/March 2009 (Fig. 6; Jentzen et al., submitted).

### 3.3 Mg/Ca-based ocean temperature assessment from living foraminifers

In order to evaluate Mg/Ca as proxy for seawater temperature, we compared Mg/Ca-temperature estimates of living specimens to (i) measured in situ temperatures and (ii) Mg/Ca-temperature estimates of fossil tests from surface sediments.

Within this study, Mg/Ca analyses were performed on bulk foraminiferal samples measured by ICP-OES and single tests measured by LA-ICP-MS. ICP-OES samples of *G. sacculifer*, *N. dutertrei* and *G. ungulata* yield higher Mg/Ca ratios on average compared to LA-ICP-MS samples from the same MSN sample (Table 3). The data indicate a difference of 0.5 ±0.5 mmol mol[-1] for *G. sacculifer* (average value of eight MSN sampling intervals), 1.2 mmol mol[-1] for *N. dutertrei* (one MSN sampling interval) and 0.17 ±0.05 mmol mol[-1] for *G. ungulata* (three MSN sampling intervals). We compare the results of

both methods to each other having in mind the data discrepancy originating from the different analytical techniques. For LA-



ICP-MS only small amounts of foraminiferal calcite from single chambers are analysed and for the ICP-MS the bulk calcite from whole foraminiferal tests are measured.

Our Mg/Ca ratios of eight species collected at specific ocean temperature ranges (corresponding to different water depth intervals) are in good agreement with established species-specific Mg/Ca-temperature calibrations (Fig. 7; cf. Supplement
S2), and further support the foraminiferal Mg/Ca-dependency on ambient water temperature. Hence, we estimate Mg/Ca-temperatures applying the best fitting calibration for each species (Fig. 8). Overall, all specimens collected in the surface waters of the eastern Gulf of Mexico (PF samples) yield low Mg/Ca-temperature estimates (averaged ~20.6 °C) according to the low early spring temperatures of ~20 °C prevailing during cruise M78/1 (Fig. 1). Higher Mg/Ca-temperature estimates (~25 °C) of shallow dwellers (symbiont and facultative symbiont bearing species) in the Florida Straits and Caribbean Sea
(MSN samples) point to higher temperatures in the mixed layer (>24 °C). Low Mg/Ca ratios of deep dwellers (*G. truncatulinoides* and *G. tumida*) in the thermocline follow the decreasing ambient seawater temperatures (Fig. 8).

### 3.3.1 (Facultative) symbiont bearing species

Our dataset is most complete for *G. sacculifer*, allowing for a detailed comparison between Mg/Ca-based temperature estimates from plankton net and surface sediment samples. In the Caribbean Sea, the estimated Mg/Ca-temperatures for
*G. sacculifer* (~26 °C) are consistent with in situ temperatures of the mixed layer (~26.2 °C), the average habitat temperature (~26 °C, derived from the standing stock, Table 2) and Mg/Ca-temperatures derived from fossil tests (~26 °C) (Fig. 8). Below 150 m water depth, the deviation between Mg/Ca-temperature and the ambient seawater temperature increases, which support the former conclusion based on $\delta^{18}O_{calcite}$ that *G. sacculifer* completed calcifying above or within the thermocline. Lower temperature estimates of ~24 °C in the Florida Straits (Station 211) (Fig. 7) mirror the generally lower sea surface
temperatures of ~24.6 °C at this station during cruise M78/1 (Fig. 2). Here the fossil tests from surface sediments yield higher Mg/Ca ratios (+0.7 mmol mol$^{-1}$) than the living specimens. The Mg/Ca-temperature of fossil specimens indicate ~26.5 °C, which is rather comparable to temperatures in the Florida Straits of the mixed layer in May (Locarnini et al., 2013, Fig. 2). Foraminiferal census data from the MSN samples suppose that the highest population density of *G. sacculifer*, consequently also the highest flux and accumulation rate of empty tests on the seafloor, appears during early spring in the
Caribbean Sea, linking this species to the warm and oligotrophic Caribbean Water (CW) (~26 °C) (Jentzen et al., submitted). Furthermore, high frequencies of *G. sacculifer* are related with the strength of the Loop Current transporting warm Caribbean Water into the Gulf of Mexico (Poore et al., 2013). Therefore, we presume that a higher flux of *G. sacculifer* in Florida Straits is likely to occur later in the year, presumably in May, hence after our sampling, and the fossil tests of *G. sacculifer* from the Caribbean Sea and Florida Straits thereby reflect different seasonal signals.
Beside the seasonal effect, millennial-scale variabilities further affect the Mg/Ca signal of fossil tests from surface sediments. Regenberg et al. (2006) assumed an age range of 2–3 kyrs in surface sediments (~0–1 cm) of the Caribbean Sea. As such, the surface sediments include the record of earlier climate variations, like the Little Ice Age, when sea surface temperatures in the Caribbean were cooler by ~2 °C (Watanabe et al., 2001). A large scatter of ~0.9 mmol mol$^{-1}$ Mg/Ca of fossil tests from Caribbean surface sediments was therefore linked partly to past environmental variabilities (Regenberg et
al., 2006). Our study, however, shows a similarly large Mg/Ca scatter in living specimens collected from the same plankton nets (MSN samples, Mg/Ca range up to ~0.87 mmol mol$^{-1}$; Fig. 7). Furthermore, LA-ICP-MS profiles across single chamber walls reveal a large Mg/Ca variability, with decreasing Mg/Ca values towards the final chamber (F) (cf. Supplement S4), which implies that "vital-effects" drive Mg$^{2+}$ incorporation. Earlier studies on surface sediments and culture experiments indicate an ontogenetic effect on the incorporation of Mg$^{2+}$ during test growth of *G. sacculifer*, with lowest Mg/Ca ratios in
the final, newly precipitated chambers (Sadekov et al., 2005; Dueñas-Bohórquez et al., 2011). Although lower average Mg/Ca ratios (~0.3 mmol mol$^{-1}$) were measured in living specimens than in fossil test, the bulk foraminiferal samples of living *G. sacculifer* from the mixed layer show a significant positive correlation between Mg/Ca and in situ temperatures



(Pearson linear, r = 0.8, p<0.05), with an overall Mg/Ca scatter comparable to that of fossil specimens from surface sediments (Fig. 7).

Our database for the other species is rather limited. Nonetheless, we can derive the following information. The symbiont bearing species *O. universa* characteristically yields very high Mg/Ca ratios in single tests (up to ~10 mmol mol$^{-1}$ on average) (cf. Lea et al., 1999; Russel et al., 2004). Mg/Ca-temperature estimates of *O. universa* are on average ~1 °C lower than the measured in situ temperature, but show decreasing values in larger depths according to lower in situ temperatures (Fig. 8; Table 3). The offset between Mg/Ca-temperatures of *P. obliquiloculata* and in situ temperatures vary from -3 °C to 9 °C. Both, *O. universa* and *P. obliquiloculata* show low and high Mg$^{2+}$ bands across single chambers of the tests (Supplement S4). Those bands are likely caused by physiological processes (Eggins et al., 2004; Kunioka et al., 2006; Sadekov et al., 2009; Spero et al., 2015) and reveal a large Mg/Ca variability in single chambers. Single LA-ICP-MS measurements of *N. dutertrei* yield lower Mg/Ca ratios than the ICP-OES measurements (Table 3). Here, the high Mg-heterogeneity in single chambers (cf. Fehrenbacher et al., 2017) probably caused the large offset between the two measuring techniques (see above). However, the average derived Mg/Ca-temperature of plankton bulk samples (~26.3 °C) at station 221 is in good agreement with the in situ temperature of the seawater at this station (~26.5 °C) (Fig. 8). The difference of 0.71 mmol mol$^{-1}$ Mg/Ca between the living and fossil bulk samples (Table 3) support the notion that adult specimens of *N. dutertrei* dwell at larger depths and continue calcifying (development of a crust; cf. Steinhardt et al., 2015; Fehrenbacher et al., 2017), as indicated by the lower δ$^{18}$O$_{calcite}$ values and smaller specimens collected in the upper mixed layer (Jentzen et al., submitted). Living specimens of *G. menardii* yield a Mg/Ca-temperature range between ~18 °C and 26.5 °C, which is larger but covers the temperature range of fossil tests (~23.2–25 °C) and the calculated average habitat temperature (~24.5 °C; Table 2) in the Florida Straits and Caribbean Sea (Fig. 8).

### 3.3.2 Symbiont barren species

In the Florida Straits both, bulk and single Mg/Ca measurements of *G. ungulata* yield temperature estimates of ~24 °C in the mixed layer and thermocline (Fig. 8) being congruent to the average habitat temperature of 23.8 °C during February/March 2009 (Table 2). The average Mg/Ca-temperature estimates of living and fossil *G. truncatulinoides* (~19 °C) mirror the average habitat temperature of ~20 °C during February/March 2009 (Fig. 8; Table 2). The deep dweller *G. tumida* shows a decreasing Mg/Ca-temperature trend from the mixed layer to the thermocline following the decreasing in situ temperature (Fig. 8). The fossil tests of *G. tumida* show higher average Mg/Ca ratios than the living individuals (Table 3) and yield higher temperature estimates. However, the Mg/Ca-temperature of fossil tests (~19 °C) represents the calculated average habitat temperature (~21.7 °C) far better than the living foraminifers, which show an offset to the prevailing in situ temperature of ~7 °C to 17 °C (Fig. 8) most likely due to variable crusting of the chambers (cf. Supplement S4).

### 3.4 δ$^{18}$O$_{seawater}$ relationship

The combination of foraminiferal δ$^{18}$O$_{calcite}$ and Mg/Ca-temperatures to estimate δ$^{18}$O$_{seawater}$ approximating paleo-salinity is a commonly accepted approach in paleoceanography (e.g. Lea et al., 2000; Schmidt et al., 2004; Nürnberg et al., 2008). Support derived from living foraminifers collected under natural conditions is still sparse. Our unique dataset on living planktic foraminifers in the mixed layer (>125 m water depth) at least allows us to test the abovementioned approach for the surface dweller *G. sacculifer* from the Caribbean Sea and Florida Straits (Fig. 9). As the δ$^{18}$O$_{seawater}$ estimates are strongly depending on both the applied δ$^{18}$O-paleotemperature equation and empirical Mg/Ca-calibration, we decided to apply the δ$^{18}$O-paleotemperature equation of Spero et al. (2003). This equation is based on *G. sacculifer* cultured in laboratory, which takes the large disequilibrium of δ$^{18}$O$_{calcite}$ in living specimens to the ambient seawater into account (Table 2). For the estimation of Mg/Ca-temperature, we applied the species-specific calibration of Regenberg et al. (2009) for *G. sacculifer* derived from fossil tests of surface sediments in the tropical Atlantic and Caribbean Sea. Our study shows that this




calibration reflects our in situ temperatures very close (Fig. 7). $\delta^{18}O_{seawater}$ estimates of *G. sacculifer* show a positive linear relationship with in situ $\delta^{18}O_{seawater}$ (r = 0.78) as well as with salinity (r = 0.77) (Fig. 9). Our study on living foraminifers hence provides compelling evidence that the combination of foraminiferal $\delta^{18}O_{calcite}$ and Mg/Ca-temperature reflecting ambient seawater properties reliably approximates the modern ocean salinity.

**4 Conclusions**

Our combined stable isotopes ($\delta^{18}O$ and $\delta^{13}C$) and Mg/Ca analyses on living planktic foraminifers, collected by MSN and PF from surface to max. 400 m water depth of the Caribbean Sea, the eastern Gulf of Mexico and Florida Straits, allow for the following conclusions:

(1) The large negative disequilibrium (between $\delta^{18}O_{calcite}$ and $\delta^{18}O_{equilibrium}$) of up to -0.35 ‰ observed for *G. sacculifer*
and *O. universa* point to a strong photosynthetic activity of the host symbionts (dinoflagellates).

(2) Ontogeny most likely controls $\delta^{18}O_{calcite}$ and $\delta^{13}C_{calcite}$ values. In this study *G. sacculifer* and *N. dutertrei* show a significant increase of $\delta^{18}O_{calcite}$ and $\delta^{13}C_{calcite}$ with increasing test size.

(3) Vertical migration in the water column and additional secretion of a calcite crust or gametogenic calcite (at the end of the foraminiferal life cycle) likely causes the increase of $\delta^{18}O_{calcite}$ with water depths and the enrichment of heavier
$^{18}O$ isotopes in fossil tests compared to living specimens.

(4) The large intraspecific scatter of Mg/Ca implies a strong "vital-effect". Nonetheless, it is evident that the ambient calcification temperature drives the Mg/Ca compositions in foraminiferal tests and causes lowered Mg/Ca derived temperature estimates at lowered in situ temperature.

(5) The various species-specific datasets agree well to published $\delta^{18}O$ and Mg/Ca calibrations.

(6) Fossil tests of *G. sacculifer* from surface sediments in the Caribbean Sea and Florida Straits suggest that the regional Mg/Ca signatures may be seasonally biased. Mg/Ca values indicate that the highest flux/accumulation rate of *G. sacculifer* occurs during spring (March) in the Caribbean Sea and delayed by a few months in the Florida Straits (most likely in May) linked to prevailing seawater temperatures of ~26 °C in the mixed layer.

(7) Combined $\delta^{18}O_{calcite}$ and Mg/Ca-temperatures of *G. sacculifer* yield $\delta^{18}O_{sewater}$ estimates, which show a positive
linear relationship with measured in situ $\delta^{18}O_{seawater}$ and salinity.

*Supplement.*
S1 Dataset
S2 Calibrations
S3 Statistics
S4 LA-ICP-MS profiles
S5 SEM Plate

*Data availability.* Dataset of this article can be found in the Supplement and in Jentzen et al. (submitted), Regenberg et al.
(2006), and Steph et al. (2009).

*Competing interests.* The authors declare that they have no conflict of interest.

*Acknowledgements.* This study was funded by the German Research Foundation DFG (grant SCHO605/8-1). The authors
thank the captain, crew and participants of RV *Sonne* cruise SO164 and RV *Meteor* cruise M78/1. We thank Nadine Gehre

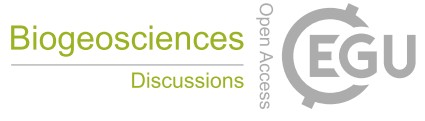

for measuring Mg/Ca on bulk samples (ICP-OES), Jan Fietzke and Steffanie Nordhausen for the help during laser ablation measurements and processing the raw data. We would like to thank Fynn Wulf and Sebastian Fessler for measuring the stable isotopes of foraminiferal calcite, Robert van Geldern (GeoZentrum Nordbayern) for measuring stable isotopes of seawater, and Birgit Mohr (Univ. Kiel) for the support with the preparation of scanning electron microscope photographs.

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



**FIGURES**

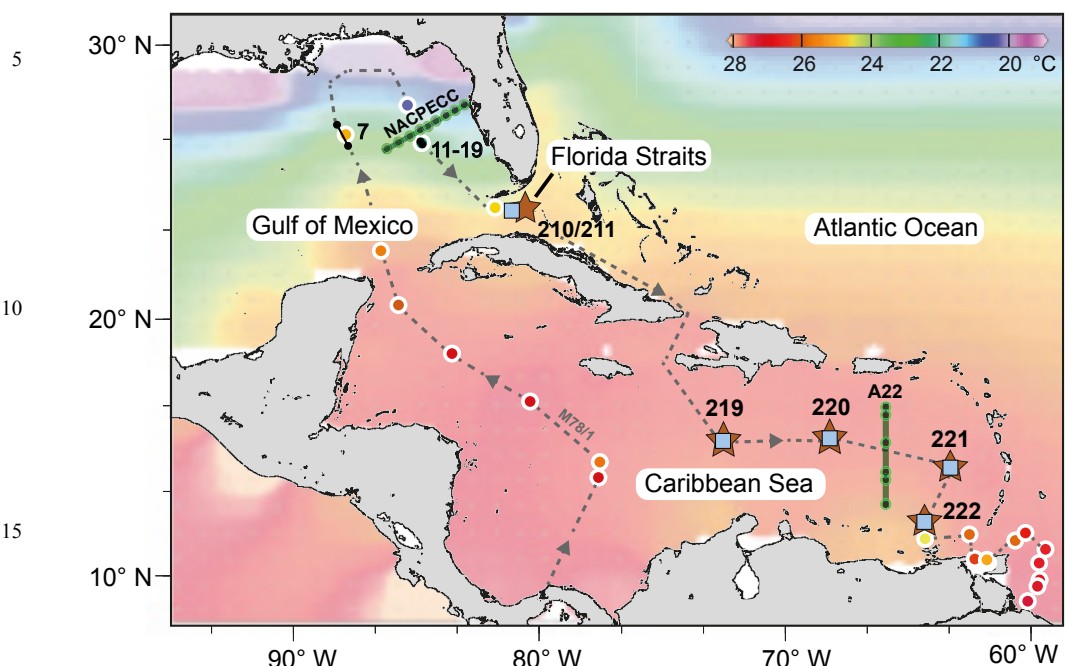

**Figure 1.** Sea surface temperature chart (SST) of the subtropical W-Atlantic (Caribbean Sea, Gulf of Mexico and Florida
Straits) showing sampling locations for living planktic foraminifers (Table 1). Brown stars: Multiclosure net samples (MSN)
and CTD stations (RV *Meteor* cruise M78/1). Black dots and lines: Plankton filter samples (PF, M78/1). Blue squares: Surface
sediment samples (M78/1 and RV *Sonne* cruise SO164, cf. Regenberg et al., 2006; Steph et al., 2009). Green lines and grey
dots: World Ocean Circulation Experiment (WOCE) transect line A22 (stations 10–15) and North American Carbon Program
(NACP) line NACPECC (stations 20–28) (cchdo.ucsd.edu). Coloured shading: SST illustrated with ODV (Schlitzer, 2009)
using World Ocean Atlas 2013 (WOA13) data from January–March (Locarnini et al., 2013). Coloured dots with white outline:
SST (3.5 m water depth) recorded during cruise M78/1 with the shipboard thermosalinograph (Schönfeld et al., 2011;
Supplement S1). Grey dashed line: Cruise track of RV *Meteor* cruise M78/1 in February and March 2009.





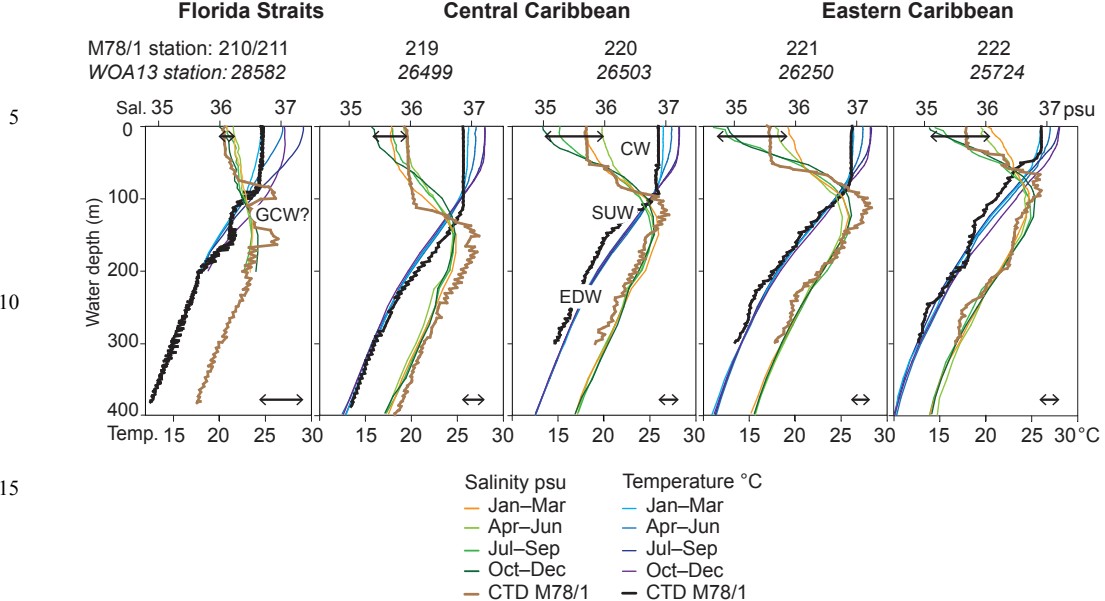

**Figure 2.** Temperature (°C) and salinity (psu) depth profiles in the working area. In situ CTD data measured during cruise M78/1 (March 2009, thick brown and black lines) are presented in comparison to the seasonally differentiated World Ocean Atlas 2013 (WOA13) data (Locarnini et al., 2013; Zweng et al., 2013; coloured thin lines). GCW: Gulf Common Water; CW: Caribbean Water; SUW: Subtropical Under Water; EDW: 18 °C Sargasso Sea Water. Black double arrows indicate the seasonal ranges of temperature (bottom) and salinity (top) in the uppermost water column (0–10 m water depth).

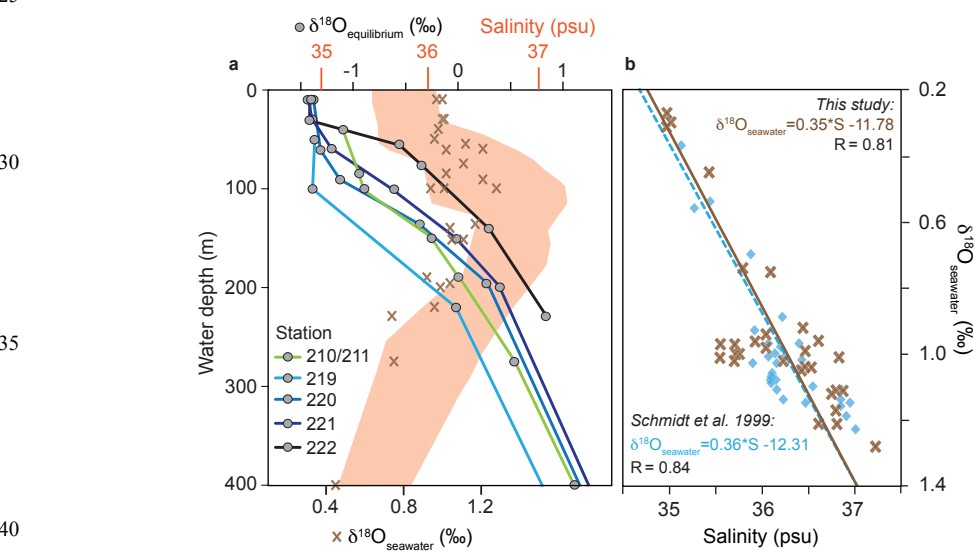

**Figure 3. a)** $\delta^{18}O_{seawater}$ (‰ VSMOW) and colour-coded $\delta^{18}O_{equilibrium}$ (‰ PDB) depth profiles at the CTD stations 210/211, 219, 220, 221, and 222 (see Fig. 1). Red shading: Salinity envelope (psu) of the ambient seawater from Florida Straits and Caribbean Sea measured during cruise M78/1 matching $\delta^{18}O_{seawater}$. **b)** Brown crosses: Measured in situ salinity vs. $\delta^{18}O_{seawater}$



in the Caribbean Sea and Florida Straits in the upper 600 meter of the water column (cf. Supplement S1 for data); blue squares: Salinity vs. $\delta^{18}O_{seawater}$ from Schmidt et al. (1999; Global Seawater Oxygen-18 Database) in the upper 600 meter of the water column in the Caribbean Sea.

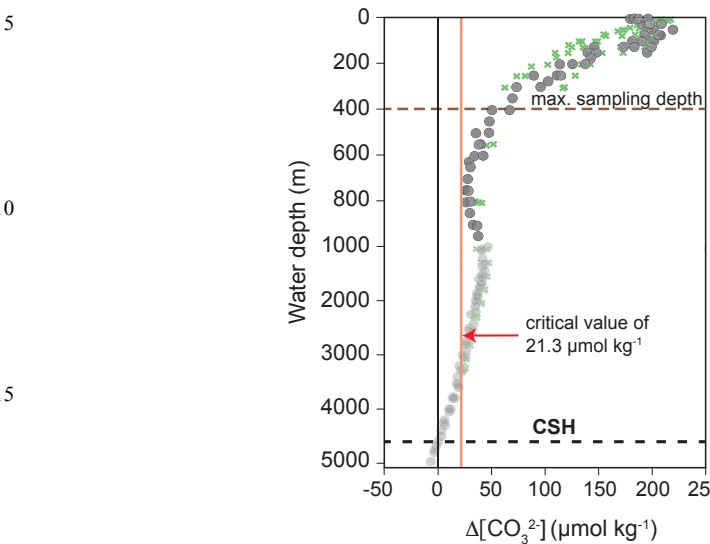

**Figure 4.** Calcite saturation state indicated by $\Delta[CO_3^{2-}]$ depth profiles of the Caribbean Sea and Gulf of Mexico. Grey dots and green crosses: Transect A22 (stations 10–15) and NACPECC (stations 20–28) (Fig. 1) with $\Delta[CO_3^{2-}]$ being the difference between $[CO_3^{2-}]_{in-situ}$ and $[CO_3^{2-}]_{saturation}$. Alkalinity and TCO$_2$ were taken from WOCE and NACP (cchdo.ucsd.edu; cruise RV *Knorr* in 1997, EXPOCODE: 316N151_4 and cruise RV *Ronald H. Brown* in 2007, EXPOCODE: 33RO20070710) to calculate $[CO_3^{2-}]_{in-situ}$ using the program CO2SYS (Pierrot et al., 2006; taking the constants ($K_1$ and $K_2$) of Mehrbach et al. (1973) refitted by Dickson and Millero (1987) and ($K_{SO4}$) from Dickson (1990)). $[CO_3^{2-}]_{saturation}$ was calculated after Jansen et al. (2002). Red vertical line indicates the critical $\Delta[CO_3^{2-}]$ value of 21.3 µmol kg$^{-1}$ below which selective Mg$^{2+}$ ion removal starts (Regenberg et al., 2014); black dashed line marks the calcite saturation horizon (CSH), which is defined to 0 µmol kg$^{-1}$ and represents the top of the lysocline at ~4600 m water depth; brown dashed line indicates the maximum plankton tow sampling depth.

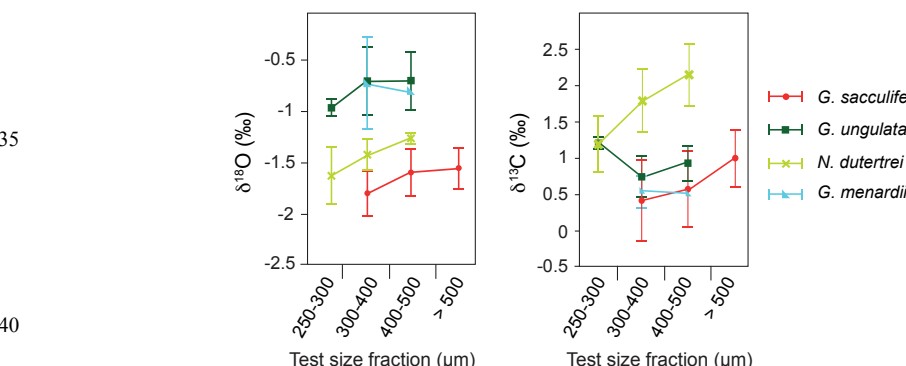

**Figure 5.** Stable oxygen and carbon isotopes (average $\delta^{18}O_{calcite}$ and $\delta^{13}C_{calcite}$ ± standard deviations) compared to different test size fractions of living planktic foraminifers (only species with more than one analysed test size fractions are depicted).





**Figure 6.** Stable oxygen isotopes of living planktic foraminifers from Florida Straits and the Caribbean Sea plotted vs. water depth (m) in comparison to calculated $\delta^{18}O_{equilibrium}$ and surface sediment data (illustrating the "vital effect"). The foraminiferal dataset was differentiated into symbiont bearing, facultative symbiont bearing, and symbiont barren species from top to bottom (Table 2; see Supplement S1 for data). Grey dots: Foraminiferal $\delta^{18}O_{calcite}$ from MSN samples, plotted at the mean sampling depth intervals. Blue shading: $\delta^{18}O_{equilibrium}$ envelope of the ambient seawater from Florida Straits and the Caribbean Sea (cf. Fig. 3a). Green bars: Range of $\delta^{18}O_{calcite}$ of fossil tests from surface sediments (green signs = average values of single stations; cf. Supplement S1). Red dashed lines: Average weighted living depths of single species during the sampling campaign in February/March 2009 (red shaded bars = the standard deviations; Table 2). Note, all test size fractions are included.



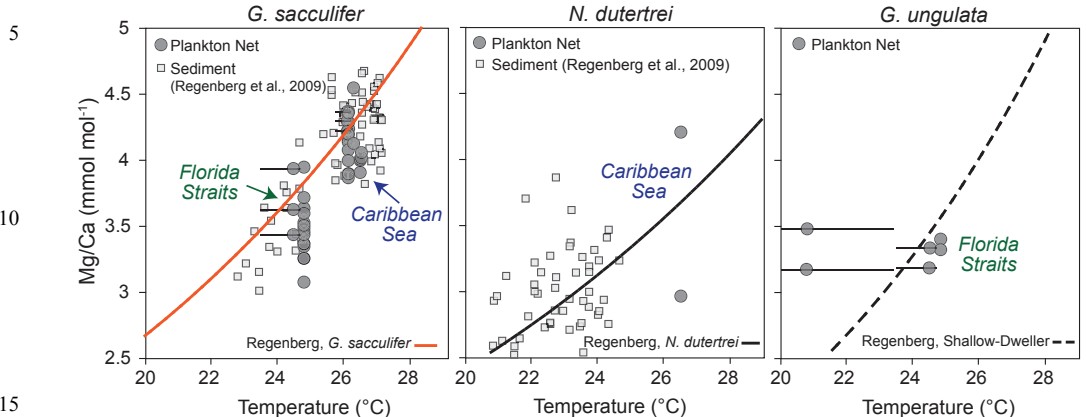

**Figure 7.** Mg/Ca values of ICP-OES bulk samples vs. temperature. Grey dots: Mg/Ca values of living specimens (*G. sacculifer*, *N. dutertrei* and *G. ungulata*), depicted at the average in situ temperature of the plankton net intervals (MSN) in the Florida Straits and Caribbean Sea recorded during cruise M78/1. Black error bars: Modern temperature ranges of the sampling intervals. Grey squares: Mg/Ca ratios of fossil tests vs. $\delta^{18}$O calcification temperature from the Caribbean Sea and tropical Atlantic modified after Regenberg et al. (2009). Orange curve: Mg/Ca calibration of Regenberg et al. (2009) (surface sediments) for *G. sacculifer*. Black curve: Mg/Ca calibration of Regenberg et al. (2009) for *N. dutertrei*. Dashed black curve: Mg/Ca calibration of Regenberg et al. (2009) for shallow dwellers.





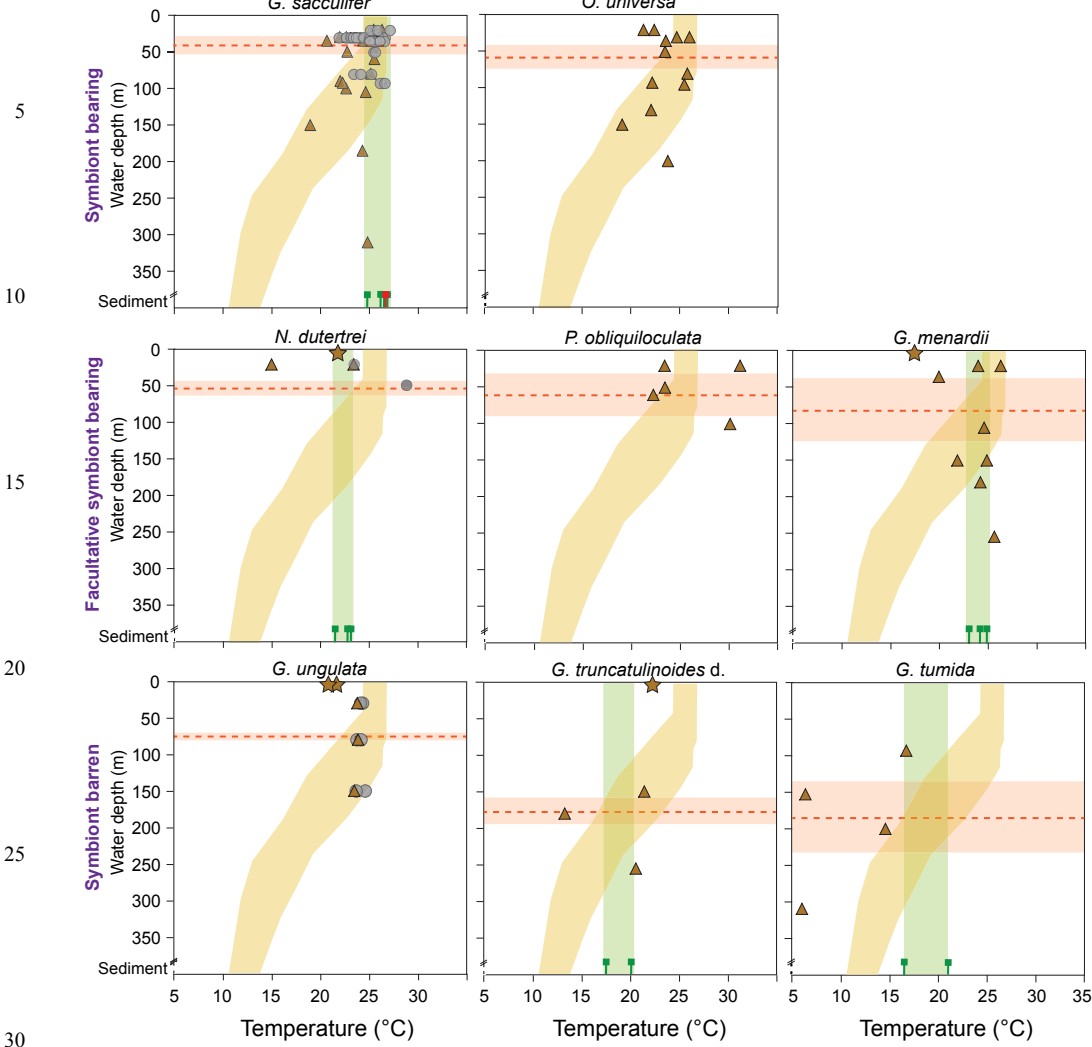

**Figure 8.** Mg/Ca derived temperature estimates of living planktic foraminifers combined from Florida Straits, the eastern Gulf of Mexico and the Caribbean Sea in comparison to the ambient seawater temperature. The foraminiferal dataset was differentiated into symbiont bearing, facultative symbiont bearing, and symbiont barren species from top to bottom (Table 2; cf. Supplement S1 for data). Grey dots: Mg/Ca-temperature estimates from bulk foraminiferal MSN samples measured by ICP-OES, depicted at the mean sampling depth intervals. Brown triangles and stars: Mg/Ca-temperature estimates derived from LA-ICP-MS measurements of single tests from MSN samples (Caribbean Sea and Florida Straits) and PF samples (Gulf of Mexico), respectively (average values, cf. Supplement S1). Yellow shading: Temperature envelope (°C) of the ambient seawater from Florida Straits and the Caribbean Sea measured during cruise M78/1 (Fig. 2; Schönfeld et al., 2011). Note: PF samples (brown stars) were taken in 3.5 m water depth in the eastern Gulf of Mexico at SST of 20 °C during cruise M78/1 (Fig. 1). Green bars: Mg/Ca derived temperature range of fossil bulk foraminiferal samples from surface sediments closest to the MSN (Green sign: Average values of single stations in the Caribbean Sea; Red sign: average value of *G. sacculifer* in the Florida Straits, cf. Supplement S1). Red dashed lines: Average weighted living depths of single species during the sampling campaign in February/March 2009 (red bars= standard deviation; Table 2). Note, all test size fractions are included.

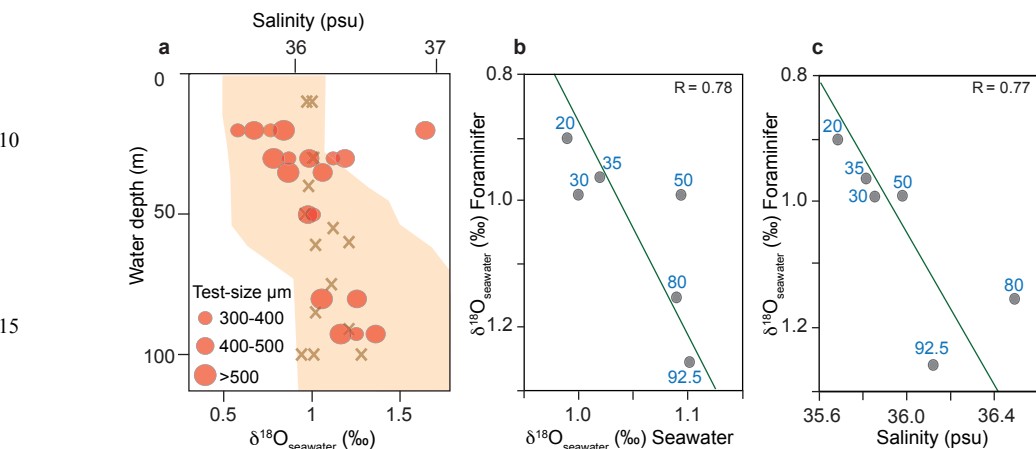

**Figure 9.** $\delta^{18}O_{seawater}$-estimates based on foraminiferal tests from living *G. sacculifer* compared to measured $\delta^{18}O_{seawater}$ and salinity recorded during cruise M78/1 in the Caribbean Sea and Florida Straits. **a)** Red dots: Average $\delta^{18}O_{seawater}$-estimates of bulk samples from different test size fractions; brown crosses: in situ $\delta^{18}O_{seawater}$ (‰ VSMOW); orange envelop: Salinity. **b)** Relationship between $\delta^{18}O_{seawater}$-estimates (foraminiferal tests) and measured $\delta^{18}O_{seawater}$ (seawater). **c)** Relationship between $\delta^{18}O_{seawater}$-estimates (foraminiferal tests) and measured in situ salinity. Grey dots indicate average values at a specific water depth (blue numbers denote average sampling water depth in m).



**TABLES**

5 **Table 1.** Station list of sediment, water and plankton samples obtained during cruises SO164 and M78/1 (Nürnberg et al., 2003; Schönfeld et al., 2011). MUC: Multicorer; GKG: Giant box corer; CTD: Conductivity Temperature Depth profiler; MSN: Hydrobios Midi multiple opening-closing plankton net; PF: Plankton filter. *indicates surface sediment sites close to MSN station (1) 219, (2) 220, (3) 221 and (4) 211 (Fig. 1).

| Cruise | Date | Device | Station No. | Latitude N (Start-End) | Longitude W (Start-End) | Water depth (m) | Sampling intervals/depth |
|---|---|---|---|---|---|---|---|
| SO164 | 27.05.2002 | MUC | 02-3 *(1) | 15°18.29 | 72°47.06 | 2977 | 0–1 cm |
| SO164 | 07.06.2002 | MUC | 22-2 *(2) | 15°24.00 | 68°12 | 4506 | 0–1 cm |
| SO164 | 09.06.2002 | MUC | 24-3 *(3) | 14°11.89 | 63°25.43 | 1545 | 0–1 cm |
| M78/1 | 10.03.2009 | MUC | 212-1 *(4) | 24°11.10 | 81°15.74 | 723 | 0–1 cm |
| M78/1 | 19.03.2009 | GKG | 222-8 | 12°1.48 | 64°28.50 | 1019 | surface |
| M78/1 | 10.03.2009 | CTD | 210-13 | 24°14.88 | 80°55.10 | 452 | 40, 85, 100, 150, 190, 275, 400 m |
| M78/1 | 10.03.2009 | CTD | 211 | 24°15.50 | 80°54.81 | 456 | - |
| M78/1 | 15.03.2009 | CTD | 219-1 | 15°18.27 | 72°47.08 | 2956 | 50, 100, 220, 600 m |
| M78/1 | 16.03.2009 | CTD | 220-1 220-2 | 15°23.99 15°23.99 | 68°12.01 68°11.99 | 4480 4480 | 10, 61, 91, 136, 196, 485 m |
| M78/1 | 18.03.2009 | CTD | 221-1 221-2 | 14°11.89 14°11.98 | 63°25.45 63°25.41 | 1534 1534 | 10, 30, 60, 100, 150, 200, 500 m |
| M78/1 | 19.03.2009 | CTD | 222-1 | 12°1.49 | 64°28.55 | 1023 | 10, 30, 55, 75, 140, 229 m |
| M78/1 | 10.03.2009 | MSN | 211-5 211-6 | 24°15.50 24°15.30 | 80°54.81 80°54.69 | 456 453 | 0–60, 60–100, 100–200, 200–300, 300–400 m |
| M78/1 | 15.03.2009 | MSN | 219-7 219-8 | 15°18.30 15°18.30 | 72°47.06 72°47.06 | 2960 2960 | 0–60, 60–125, 125–180, 180–220, 220–400 m |
| M78/1 | 17.03.2009 | MSN | 220-8 220-9 | 15°23.99 15°23.99 | 68°12.00 68°12.00 | 4481 4482 | 0–70, 70–110, 110–150, 150–220, 220–300 m |
| M78/1 | 18.03.2009 | MSN | 221-7 221-8 | 14°11.89 14°11.89 | 63°25.43 63°25.43 | 1533 1535 | 0–40, 40–60, 60–150, 150–210, 210–300 m |
| M78/1 | 19.03.2009 | MSN | 222-6 222-7 | 12°1.57 12°1.55 | 64°28.80 64°28.80 | 1031 1028 | 0–40, 40–80, 80–120, 120–180, 180–300 m |
| M78/1 | 03.03.2009 | PF | 7 | 26°31.38–27°39.86 | 87°5.32–88°16.23 | - | 3.5 m |
| M78/1 | 06.03.2009 | PF | 11 | 26°18.35–26°12.21 | 84°44.97–84°41.92 | - | 3.5 m |
| M78/1 | 06.03.2009 | PF | 12 | 26°10.7–26°12.48 | 84°44.08–84°43.40 | - | 3.5 m |
| M78/1 | 07.03.2009 | PF | 19 | 26°12.18–26°12.18 | 84°43.87–84°43.87 | - | 3.5 m |





**Table 2.** Average weighted living depth (m), habitat temperature (°C), symbionts information and $\delta^{18}O_{disequilibrium}$ values of single species from this study and other authors.

| | | G. sacculifer | N. dutertrei | O. universa | P. obliquiloculata | G. ungulata | G. menardii | G. truncatulinoides | G. tumida |
|---|---|---|---|---|---|---|---|---|---|
| **Species** | | | | | | | | | |
| **Avg. living depth (m)[x]** | | 41±9 | 54±10 | 58±16 | 61±29 | 75±5 | 81±43 | 176±18[d] | 185±49 |
| **Avg. habitat temperature (°C)[y]** | | 25.9 | 25.11 | 25.13 | 25.61 | 23.81 | 24.47 | 20.14[d] | 21.72 |
| **Symbionts** | | Dinoflagellates[1] | Chrysophycophyte[1F] | Dinoflagellates[1] | Chrysophycophyte[1F] | None?[2] | Chrysophycophyte[1F] | None[1] | None[3] |
| **Disequilibrium values** $\delta^{18}O_{calcite} - \delta^{18}O_{equilibrium}$ (‰) | Mixed layer* | -0.35 | -0.14 | -0.32 | +0.18 | +0.08 | -0.05 | | |
| | Thermocline* | -0.98 | -0.79 | -0.72 | -0.54 | +0.01 | -0.54 | +0.02[d] | -0.65 |
| | Bouvier-Soumagnac and Duplessy, 1985 | | -0.3 | -0.2 | | | -0.3 | | |
| | Duplessy et al., 1981a | -0.6 | | | | | | | |
| | Erez and Honjo, 1981[#] | -0.15 | -0.21 | -0.14 to -0.02 | -0.05 | | | -0.15 to +1.28 | |
| | Vergnaud-Grazzini, 1976 | -0.6 | | -1.0 | | | | -0.4[+] to -0.6 | |
| | Kahn, 1979 | -0.38 | -1.57 to -0.29[+] | -0.95 | | | -0.24 to 0[+] | 0[d] / -0.1 to +0.16[d+] | <0 |
| | Lončarić et al., 2006 | -0.36 to -0.03 / -0.13 to -0.16[+] | | | | | | | |
| | Shackleton et al., 1973 | -0.39 | -0.11 | | +0.06 | | | | |

\* This study (average values); × Jentzen et al (submitted); d= *G. truncatulinoides* dextral
1= Gastrich, 1987; 2= Bé, 1977; 3= Kučera, 2007
F= Facultative symbionts
+= large/thick specimens; # = seasonal variations



**Table 3.** Average values of $\delta^{18}O_{calcite}$ and Mg/Ca (measured on ICP-OES* and LA-ICP-MS) from the mixed layer, thermocline and surface sediment (cf. Supplement S1 for data). PF samples are not included in the calculations.

| Species | $\delta^{18}O_{calcite}$ (‰) | | | Mg/Ca (mmol mol$^{-1}$) | | |
|---|---|---|---|---|---|---|
| | Mixed layer | Thermocline | Sediment | Mixed layer | Thermocline | Sediment |
| *G. sacculifer* | -1.62 | -1.52 | -1.38 | 3.87*/ 3.51 | 3.52 | 4.20* |
| *P. obliquiloculata* | -1.15 | -1.07 | -0.55 | 2.84 | 2.86 | |
| *O. universa* | -1.53 | -1.13 | -1.15 | 8.33 | 7.61 | |
| *N. dutertrei* | -1.51 | -1.37 | -0.4 | 3.59*/ 2.36 | | 2.88* |
| *G. ungulata* | -0.95 | -0.26 | -0.67 | 3.30*/ 3.20 | 3.32*/ 3.10 | |
| *G. menardii* | -1.01 | -0.73 | -0.54 | 3.10 | 3.19 | 3.27* |
| *G. tumida* | | -0.58 | -0.11 | 2.45 | 1.80 | 2.68* |
| *G. truncatulinoides* d. | | 0.28 | 1.13 | | 2.5 | 2.52* |
| *Bulk samples (measured on ICP-OES) | | | | | | |