# Peer review of "Mg/Ca and $\delta^{18}$ O in living planktic foraminifers from the Caribbean, Gulf of Mexico and Florida Straits"

_Biogeosciences, 2018_

## Referee Comment (RC1) · T. Toyofuku (Referee) · 6 Jun 2018

T. Toyofuku (Referee)

toyofuku@jamstec.go.jp

In this study, eight planktonic foraminifera are collected from water mass or sediment, and the authors investigate how Mg / Ca and d18O of the shell reflects the water mass property. Eight species are big numbers. The reviewer has considered it is ambitious project to comprehensively study Mg / Ca and d18O of planktonic foraminifera in the Caribbean waters. This theme is a very interesting result for paleoceanographers who commit paleoenvironmental analysis using planktonic foraminifers and micropaleontologists/geochemists who study trace elements and isotopic composition of foraminiferal calcite. In recent years, laboratory cultures have reveaed how trace elements and isotope compositions of planktonic foraminifera are distributed by growing environments. This study can also be read as an answer paper from field studies for laboratory culture trials. Reviewer could conclude it is a suitable manuscript to publish on Biogeosciences.

According to recent molecular phylogenetic studies have revealed that morphological species can be divided into sub groups. Even no genomic analysis is necessary in this study, but materials to identify the morphological species used in this study are necessary. By showing the SEM plate as a supplement, compatibility with other studies is maintained. Measurement methods and their limitations are clearly shown and this reviewer could not find any problems. Further, manuscripts are well prepared and high quality.

Question

The authors pointed out the importance of carbonate ion concentration in the introduction section (P1L33) and show the vertical profile of carbonate ion concentration (Fig. 4), although the influence of carbonate ion concentration on neither oxygen/carbon isotopic ratio nor Mg/Ca has not been discussed. The carbonate ion concentration profile shows the most steep change in the surface layer from 0-400m, but perhaps this variability did not affect the paleoenvironmental analysis with core-top samples? In fact, the authors compare between living samples and fossil samples, which are in good agreement. Perhaps, in the field samples, fluctuations by carbonate ion concentration have already been incorporated, so does that mean it will work well without consideration about carbonate ion concentration? Since reviewers are worked through laboratory culture experiments, the factors that pH, carbonate ion concentration fluctuations, etc. in addition to water temperature and salinity influence the shell element / isotope composition can not be ignored. Under what conditions does carbonate ion concentration need not be taken into consideration?

Minor comment

The authors can include Zeebe, Richard E. Geochimica et Cosmochimica Acta 63.13-14 (1999): 2001-2007 was good because it is an important literature as well as Spero

et al. (1997) and Bijma et al. (1999).

Definition of Vital effects In this research, the reviewer highly appreciated that the vital effect is finely verified. Is it possible to explicitly specify the Vital effect assumed by this research in an introduction or section 3.2 by dividing it into elements (eg ontogenic effect, symbiotic effect, calcification depth, ecology including optimal season / annual cycle etc)? The authors discuss about the influence of each of these factors, but it is rather enumerative.

---

## Referee Comment (RC2) · B. Metcalfe (Referee) · 3 Jul 2018

Brett Metcalfe (LSCE, FR & VUA, NL)

Jentzen et al. have performed extensive stable isotope and trace metal geochemical analysis on 8 species from samples from 5 multinets, 4 plankton filter and 5 core tops. This is a vast dataset that merits publication, however, the paper could be improved in several ways. The first is to plot the d18O and Mg/Ca for individual stations so readers can see which datapoint goes with which station. Second the Mg/Ca-d18O and Mg/Ca data could be toned down a bit as it is a bit over sold, the high r value of the Mg/Ca-d18O data (Figure 9) is a misnomer and the measured salinity and d18Osw are not reconstructed using foraminifera accurately enough for palaeoclimate reconstructions.

For instance, how would pooling of the various size fractions into a single mean for Mg/Ca-d18O (Figure 9) work practically in a proxy study down-core? Should shells >300 um be pooled and measured to reduce the 'noise'? The paper is nicely written, with a valid dataset, the results are interesting and should be expressed a bit more openly (i.e. less focus on the combined plots and more on the individual stations). These comments are outlined in more detail below.

Major Comments

(1) Raw data. My major problem with the paper is the lack of presentation of the 'raw' station data, figure's 6 and 8 are a synthesis of the entire dataset which whilst interesting should not be how the reader sees the data. There is a 4 page table of $\delta$18O measurements and 3 pages of Mg/Ca which is a lot of work that the authors have done, that shouldn't just be in table format in the supplement! I would like to see the data plotted per station (I think in the main text, but also could be in the supplement though it might get overlooked) so that the reader can see how the various stations/species isotope values 'evolve'. For instance, a figure (5x8 panel) of the 5 multinet stations depth T, $\delta$18O, and salinity profiles with the values of each species in a different panel. Or the authors could extend figure 2 to include the isotopes/trace metal geochemical values. I understand that these plots also include filter and sediment values which may explain the rationale of the authors for plotting the data like it is.

(2) Mg/Ca-d18O. Does it really work? The authors show that they have a r value of 0.78 and 0.77 for d18Osw and salinity observed vs expected and the results in Figure 9 gives compelling evidence for the use of Mg/Ca-d18O to estimate d18Osw. In figure 9a the range in d18O estimates is more than 0.5 per mil but its difficult to tell whether this could be due to the spread in the station data. As per my previous comment I believe you need to show the individual station data, as it's impossible to link the salinity, d18Osw and estimates from individual stations to one another.

The authors should check the significance of the r values, for n = 6 (degrees of freedom

are n-2) the significance value of r at an alpha levels of 0.5 and 0.1 is 0.812 and 0.917 respectively. It would also help to propagate the errors between the two analytical measurements.

Furthermore, Figure 9b, the scale of the y and x axis differ considerably, the entire data's x-axis range is approximately one tick along the y-axis. It would also appear that were the authors to draw a 1:1 line it would be offset from the line the authors have drawn through the data. Combining these two points it appears there is a weak relationship that isn't statistically significant and the approach doesn't exactly predict the 'real values'.

In addition, in Figure 9b it would appear that only one axis is 'reversed', the current view of the plot, at a glance without looking at the absolute values along either axis, gives the impression of a negative correlation instead of the positive correlation that it is and the authors state (Pg. 10 Line 1).

Why not plot the Figure 3b line (d18Osw vs salinity) in Figure 9c? Just from eye-balling it I would say that the slopes are different. Or convert the d18Osw into salinity (though I will admit that might impose some circular reasoning)?

I would say that the authors data and perhaps this approach doesn't provide accurate d18Osw estimates. Nor do they elaborate upon the influence of test-size upon their estimates, how will this be influenced if foraminiferal size is not static through time (Peeters et al., 1999 Mar Micro; Metcalfe et al., 2015 Biogeosciences) and if this result is dependent upon pooling different size fraction (>300 um) measurements.

(3) Mg/Ca. First the amazing supplementary figures: S2 Figure 1 and Figure 2 would be better in the text than Figure 7, however these figures show that there is a lot of scatter within the data generated that should be elaborated upon in the text. I disagree with the authors that (Pg. 8 Lines 3 – 5): "Our Mg/Ca ratios of eight species collected at specific ocean temperature ranges (corresponding to different water depth intervals) are in good agreement with established species-specific Mg/Ca temperature calibrations (Fig

7 cf. Supplement intervals) and further support the foraminiferal Mg/Ca-dependency on ambient water temperature". The plot of G. menardii in S2 Figure 2 could be best described as 'shotgun' like; G. tumida is recording 5oC (Figure 8) and G. ungulata appears to be getting warmer with depth (Figure 8). The authors themselves state: (Pg. 9 Line 7) "the offset between. . .vary from -3C to 9C"; (Pg. 9 Line 28) "the Mg/Ca temperature of fossil tests ($\sim$19C) represent the calculated average habitat temperature ($\sim$21.7C) far better than the living foraminifers". But this is not a bad thing, this is the data and the authors should present it a bit differently (e.g. Pg 10 Line 19 "datasets agree well to published d18O and Mg/Ca calibrations", do they?if these are species specific calibrations should they have offsets?).

I think the authors should add an x-axis error bar in S2 Figure 2, but also consider that collected foraminifer may not have actually calcified in the collected interval so perhaps extend this error bar to incorporate the temperature of shallower depths. Such an approach might explain some of the high Mg/Ca values in the lower temperatures as specimens that had yet to calcify in the water they were caught in. Could this explain the discrepancy with recorded temperatures? One assumption the authors have made is that living foraminifera caught in a net interval are calcifying in it, for filter and the shallowest net it can be assumed that the bulk of the shell (considering that some foraminifera could ascend to the surface during their juvenile stage) comes from that net interval. This assumption doesn't hold true for the deeper nets.

It would be interesting to consider the difference between living and dead shell geochemistry (in this or another paper), and whether by choosing only living foraminifera the results could be biased (just because it is alive doesn't mean it has to represent the values it was caught in). Highlighting the shell concentration for each net interval could indicate whether the deeper depths are shells of living foraminifera sinking outside of their habitat zone but still alive (see Peeters et al., 2002; Global and Planetary Change – for examples e.g. Figures 5 and 6). This is alluded to by the dashed red-line in Figure 8, which shows the weighted average living depths shallower than some of

the measured depth intervals for some species.

Minor Comments

Check the plural and singular of the word foraminifera throughout the text, as well as V-PDB and V-SMOW (sometimes it's PDB)

Pg. 1 Line 8 (first line of Abstract): The first line (slightly) contradicts the second line (if it is successfully approximated, why is refinement needed?) perhaps change 'are successfully' to 'can be' and add 'with varying success' to the end of the sentence

Pg. 1 Line 15 add 'with respect' between disequilibria and to

Pg. 1 Line 36: Avoid starting a sentence with Mg/Ca change to "The ratio of Mg/Ca"

Pg. 2 Line 5: add 'for proxy users' after critical

Pg. 2 Line 7: Perhaps a re-wording? Whilst, relatively few (isotope) geochemical studies have been conducted on recent/living planktic foraminifers, either collected from the water column or culture under controlled conditions, these studies are important for assessing different controlling factors on d18Ocalcilte and Mg/Ca during biomineralization.

Pg. 2 Line 7: Also 'relatively few'? I would disagree with few d18O studies, though Mg/Ca maybe.

Pg. 2 Line 12: 'Here we' instead of 'We here'

Pg. 2 Line 21 move 'and fossil' to the end of the line (so it reads 'below the ship, and fossil foraminifera from sediments')

Pg. 2 line 32 – 34: Why poison d18O samples? Does the addition of poison impact cavity-ring down / infrared spectroscopy, as there is some suggestion in the literature that salinity may have an impact on the vaporiser unit (especially changing salinity between samples but also certain salt solutions)

Pg. 2 line 29/Pg. 3 line 1: Bradshaw (1957) did this massive plankton net study/database of the Pacific, halfway through though he stopped using Rose Bengal to identify 'living' as cytoplasm in the shell was just as efficient. Out of curiosity, is there a reason why (Pg. 3 Line 1) 'cytoplasm-bearing' was picked rather than stained? Is there any potential analytical error associated with staining?

Pg. 2 line 40: perhaps add 'trilobus-like' to better describe forms of T. sacculifer with a 'spherical last chamber'.

Pg. 2 line 41: Globorotalia ungulata, the arch that distinguishes it from G. menardii did you perchance consider laser ablating the arch and the rest of the chamber separately? More out of curiosity, it looks like a chamber feature rather than an embellishment (like a keel) so I wonder is it identical to the rest of the chamber but thicker? Or does it have some structural modification that can be obsered in the ablation profile.

Pg. 2 line 41: Globorotalia truncatulinoides dextral, have you seen the new paper by Reynolds et al (2018; Mar. Micro.)? Did you distinguish between crusted and encrusted G. truncatulinoides? Is there any ablation data that might shed light on the Mg/Ca content of encrusted and non-encrusted from your data?

Pg. 3 Lines 1-4. Add in size fraction used for analysis (from the table it appears not all size fractions picked on pg. 2 line 37 were included in the d18O analysis).

Pg. 3 Line 14. The magazine/turret of a Kiel device is quite small, perhaps give an indicator of the number of standards per run or in total measured? (as per the Mg/Ca analysis i.e. Pg. 4 Line 22)

Pg. 3 Line 18. Add Picarro before model (I assume it's a Picarro instrument based upon L1102-i).

Pg. 3 Line 26. Whose rearrangement of Kim and O'Neil?

Pg. 3 Line 31. Add in a mean number of specimens with a plus/minus.

Pg. 4 Line 2. The steps were repeated several (1-2) times to completely remove the cytoplasm, could varying this step have any impact on the resultant values between samples that have had a single step and a replicate step?

Pg. 4 Line 21. Outliers are classified as those values that fall 2 stdev above or below the mean, but that would mean based upon the proportion without (assuming normality) that ~4.55% of the data could be removed. What's the rationale for this? How much does this influence the data?

Pg. 6 Line 18. The transition from juvenile to neanic to adult stages occurs between 100 and 200 um (Brummer et al., 1986 Mar Micro; 1987 Nature), therefore foraminifera above >200 um (species dependent) should be considered as 'adult'. Could the vital effect, described here, instead be due to growth rate (adult specimens with different sized chambers could suggest variation in growth).

Pg. 7 Line 1. Regarding GAM, if you assume that a shell is a weighted average, if 25% of the shell is composed of gam-calcite and it has a 0.5 per mil offset from living, if you consider it proportionally the gam-calcite would need to have a significantly larger offset from the rest of the living shell.

Pg. 7 Line 34. Would it not be appropriate to (like the palaeotemperature d18O eq1) compare the expected Mg/Ca value by calculating an expected Mg/Ca value (assuming that Mg and Ca concentrations are not limiting and using the values in S2 Table 2)

Figure 1. It's a beautiful figure. However, two things, (1.) the map has a transparency that the scale bar does not, could make it difficult to interpret the color, (2.) The rainbow is a bit difficult for colour blind readers to see (I am guilty of this as well). See: https://www.climate-lab-book.ac.uk/2014/end-of-the-rainbow/

Figure 3. Why not plot the figure like Figure 2 (with isotope, salinity and equilibrium plotted per station) rather than as a composite plot?

Figure 9. (see comments above)

[Figure]

---

## Author Comment (AC1) · 23 Sep 2018

We kindly thank Takashi Toyofuku for the positive feedback, very constructive and thoughtful comments, which greatly helped to improve our manuscript. Below you find our responses to each point which was addressed and how we incorporated all suggestions in our revised manuscript. Referee comments are written in italics and the respective answers are given in normal font in blue.

Yours sincerely

Anna Jentzen and on behalf of all co-authors

**Answers to referee 1:** *T. Toyofuku*

*Question:*
*The authors pointed out the importance of carbonate ion concentration in the introduction section (P1L33) and show the vertical profile of carbonate ion concentration (Fig. 4), although the influence of carbonate ion concentration on neither oxygen/carbon isotopic ratio nor Mg/Ca has not been discussed. The carbonate ion concentration profile shows the most steep change in the surface layer from 0-400m, but perhaps this variability did not affect the paleoenvironmental analysis with core-top samples? In fact, the authors compare between living samples and fossil samples, which are in good agreement. Perhaps, in the field samples, fluctuations by carbonate ion concentration have already been incorporated, so does that mean it will work well without consideration about carbonate ion concentration? Since reviewers are worked through laboratory culture experiments, the factors that pH, carbonate ion concentration fluctuations, etc. in addition to water temperature and salinity influence the shell element / isotope composition cannot be ignored. Under what conditions does carbonate ion concentration need not be taken into consideration?*
Answer: We agree to the referee that the effect of the carbonate ion concentration on foraminiferal tests has not been discussed enough in our first version. We added further information on this topic (including suggested literature) in Chapter 2.5 (P5L2 to P5L17). We don't think that the carbonate ion concentration can be ignored and we believe that it plays an important role for the interpretation of element/isotope compositions of foraminiferal tests. This is the reason why we use the corrected Mg/Ca values for fossil tests from the sediments below 2500 m water depth, according to Regenberg et al. (2009), based on the study of Regenberg et al. (2006, Effect of $\Delta[CO_3^{2-}]$ on foraminiferal tests from the Caribbean Sea.)

*Minor comments:*
*The authors can include Zeebe, Richard E. Geochimica et Cosmochimica Acta 63.13- 14 (1999): 2001-2007 was good because it is an important literature as well as Spero et al. (1997) and Bijma et al. (1999).*
Answer: We thank the referee for the suggested literature and we added it to our manuscript.

*Definition of Vital effects in this research, the reviewer highly appreciated that the vital effect is finely verified. Is it possible to explicitly specify the Vital effect assumed by this research in an introduction or section 3.2 by dividing it into elements (eg ontogenic effect, symbiotic effect, calcification depth, ecology including optimal season / annual cycle etc)? The authors discuss about the influence of each of these factors, but it is rather enumerative.*
Answer: We thank the referee for this comment on the vital effect, which we now discuss in more detail in the introduction (P1L35–L37; P2L9–L12).

[revised manuscript text omitted]

---

## Author Comment (AC2) · 23 Sep 2018

We kindly thank Brett Metcalfe for the positive feedback, very constructive and thoughtful comments, which greatly helped to improve our manuscript. Below you find our responses to each point which was addressed and how we incorporated all suggestions in our revised manuscript. Referee comments are written in italics and the respective answers are given in normal font in blue.

Yours sincerely

Anna Jentzen and on behalf of all co-authors

**Answers to referee 2:** *B. Metcalfe*

*Major Comments:*
*1. Raw data. My major problem with the paper is the lack of presentation of the 'raw' station data, figure's 6 and 8 are a synthesis of the entire dataset which whilst interesting should not be how the reader sees the data. There is a 4 page table of δ18O measurements and 3 pages of Mg/Ca which is a lot of work that the authors have done, that shouldn't just be in table format in the supplement! I would like to see the data plotted per station (I think in the main text, but also could be in the supplement though it might get overlooked) so that the reader can see how the various stations/species isotope values 'evolve'. For instance, a figure (5x8 panel) of the 5 multinet stations depth T, δ18O, and salinity profiles with the values of each species in a different panel. Or the authors could extend figure 2 to include the isotopes/trace metal geochemical values. I understand that these plots also include filter and sediment values which may explain the rationale of the authors for plotting the data like it is.*
Answer: We thank the referee for this comment. However, we think the presentation of the raw data for each station is too much within the main text, which is the reason why we plot the data for each species together. Anyway, we agree that some information is "lost" and therefore we provide some detailed plots in the new supplement S6.

*2. Mg/Ca-d18O. Does it really work? The authors show that they have a r value of 0.78 and 0.77 for d18Osw and salinity observed vs expected and the results in Figure 9 gives compelling evidence for the use of Mg/Ca-d18O to estimate d18Osw. In figure 9a the range in d18O estimates is more than 0.5 per mil but its difficult to tell whether this could be due to the spread in the station data. As per my previous comment I believe you need to show the individual station data, as it's impossible to link the salinity, d18Osw and estimates from individual stations to one another. The authors should check the significance of the r values, for n = 6 (degrees of freedom are n-2) the significance value of r at an alpha levels of 0.5 and 0.1 is 0.812 and 0.917 respectively. It would also help to propagate the errors between the two analytical measurements. Furthermore, Figure 9b, the scale of the y and x axis differ considerably, the entire data's x-axis range is approximately one tick along the y-axis. It would also appear that were the authors to draw a 1:1 line it would be offset from the line the authors have drawn through the data. Combining these two points it appears there is a weak relationship that isn't statistically significant and the approach doesn't exactly predict the 'real values'. In addition, in Figure 9b it would appear that only one axis is 'reversed', the current view of the plot, at a glance without looking at the absolute values along either axis, gives the impression of a negative correlation instead of the positive correlation that it is and the authors state (Pg. 10 Line 1). Why not plot the Figure 3b line (d18Osw vs salinity) in Figure 9c? Just from eye-balling it I would say that the slopes are different. Or convert the d18Osw into salinity (though I will admit that might impose some circular reasoning)? I would say that the authors data and perhaps this approach doesn't provide accurate d18Osw estimates. Nor do they elaborate upon the influence of test-size upon their estimates, how will this be influenced if foraminiferal size is not static through time (Peeters et al., 1999 Mar Micro; Metcalfe et al., 2015 Biogeosciences) and if this result is dependent upon pooling different size fraction (>300 um) measurements.*

Answer: We thank the referee for all the remarks above. However, we think that the Mg/Ca-$\delta^{18}$O relationship does work, even though we admit to be more careful in our interpretation. We add two new Figures: 9c and 9d (see below) showing the relationship between estimated $\delta^{18}$O and measured $\delta^{18}$O and salinity for each station. Furthermore, we calculated the p values for the average dataset (n=6; degree of freedom 4), which results in p values <0.08 for both relationships (Fig. 9b). This indicates that we have no statistically significant relationship. One reason might be the small number of data, which don't have a large salinity/$\delta^{18}$O$_{seawater}$ range. The different scale between the x and y axis does not influence the outcome of the relationship, however we plot a 1:1 scale for Fig. 9c and 9d. Fig. 3b shows the relationship between salinity and $\delta^{18}$O$_{seawater}$ of the upper 600 m of the water column, which is not the same range as for *G. sacculifer*. On the last point, we agree on the effect of the test-size (statistically significant for $\delta^{18}$O of *G. sacculifer* see Fig. 5). Unfortunately, we do not have enough data for all test-sizes to give further information on this very important topic.

[Figure]

NEW: Figure 9: c) Relationship between $\delta^{18}$O$_{seawater}$-estimates (foraminiferal tests) and measured $\delta^{18}$O$_{seawater}$ (seawater) for each station. d) Relationship between $\delta^{18}$O$_{seawater}$ -estimates (foraminiferal tests) and measured salinity for each station.

*3. Mg/Ca. First the amazing supplementary figures: S2 Figure 1 and Figure 2 would be better in the text than Figure 7, however these figures show that there is a lot of scatter within the data generated that should be elaborated upon in the text. I disagree with the authors that (Pg. 8 Lines 3 – 5): "Our Mg/Ca ratios of eight species collected at specific ocean temperature ranges (corresponding to different water depth intervals) are in good agreement with established species-specific Mg/Ca temperature calibrations (Fig 7 cf. Supplement intervals) and further support the foraminiferal Mg/Ca-dependency on ambient water temperature". The plot of G. menardii in S2 Figure 2 could be best described as 'shotgun' like; G. tumida is recording 5oC (Figure 8) and G. ungulata ap- pears to be getting warmer with depth (Figure 8). The authors themselves state: (Pg. 9 Line 7) "the offset between. . .vary from -3C to 9C"; (Pg. 9 Line 28) "the Mg/Ca temperature of fossil tests (~19C) represent the calculated average habitat temperature (~21.7C) far better than the living foraminifers". But this is not a bad thing, this is the data and the authors should present it a bit differently (e.g. Pg 10 Line 19 "datasets agree well to published d18O and Mg/Ca calibrations", do they?if these are species specific calibrations should they have offsets?). I think the authors should add an x-axis error bar in S2 Figure 2, but also consider that collected foraminifer may not*

*have actually calcified in the collected interval so perhaps extend this error bar to incorporate the temperature of shallower depths. Such an approach might explain some of the high Mg/Ca values in the lower temperatures as specimens that had yet to calcify in the water they were caught in. Could this explain the discrepancy with recorded temperatures? One assumption the authors have made is that living foraminifera caught in a net interval are calcifying in it, for filter and the shallowest net it can be assumed that the bulk of the shell (considering that some foraminifera could ascend to the surface during their juvenile stage) comes from that net interval. This assumption doesn't hold true for the deeper nets. It would be interesting to consider the difference between living and dead shell geo- chemistry (in this or another paper), and whether by choosing only living foraminifera the results could be biased (just because it is alive doesn't mean it has to represent the values it was caught in). Highlighting the shell concentration for each net interval could indicate whether the deeper depths are shells of living foraminifera sinking out- side of their habitat zone but still alive (see Peeters et al., 2002; Global and Planetary Change – for examples e.g. Figures 5 and 6). This is alluded to by the dashed red-line in Figure 8, which shows the weighted average living depths shallower than some of the measured depth intervals for some species.*

Answer: S2 Figure 1 and 2 do not show exactly our interpretation in the main text. We do not believe that all foraminifers precipitate their tests in the sampling depth where we collected the individuals (see *G. sacculifer* P8 L26–27; P6L33–38). This is exactly the reason why we have higher Mg/Ca temperatures estimates for some specimens. For example: Mg/Ca values for *G. menardii* fit pretty well the Mg/Ca estimates from the sediment and the calculated average habitat depth, however, we have high Mg/Ca values of specimens deeper in the water column (>250 m water depth; Fig. 8). Therefore, we can assume that *G. menardii* does not calcify deeper in the water column than 250 m water depth, even though we found a specimen alive (we add this assumption in our manuscript P9L30). The species *G. ungulata* has an increase of Mg/Ca of 0.02 mmol/mol (bulk samples see Table 3), which is smaller than the analytic precision. We point out that specimens of *G. tumida*, which are collected in the deeper water column, have a variable crust (see P9L41), which lowers Mg/Ca values. Furthermore, most of our living foraminifers did not yet finish their life cycle, which results in different Mg/Ca values. Counting all these aspects together, we believe our data agree well with existing calibrations and give reasonable Mg/Ca values.
We added an x-axis error bar in S2 Figure 2.

**Minor Comments**

*Check the plural and singular of the word foraminifera throughout the text, as well as V-PDB and V-SMOW (sometimes it's PDB)*
Done

*Pg. 1 Line 8 (first line of Abstract): The first line (slightly) contradicts the second line (if it is successfully approximated, why is refinement needed?) perhaps change 'are successfully' to 'can be' and add 'with varying success' to the end of the sentence*
Done

*Pg. 1 Line 15 add 'with respect' between disequilibria and to*
Done

*Pg. 1 Line 36: Avoid starting a sentence with Mg/Ca change to "The ratio of Mg/Ca"*
Done

*Pg. 2 Line 5: add 'for proxy users' after critical*
Done

*Pg. 2 Line 7: Perhaps a re-wording? Whilst, relatively few (isotope) geochemical stud- ies have been conducted on recent/living planktic foraminifers, either collected from the water column or culture under*

*controlled conditions, these studies are important for assessing different controlling factors on d18Ocalcilte and Mg/Ca during biominer- alization.*
Done

*Pg. 2 Line 7: Also 'relatively few'? I would disagree with few d18O studies, though Mg/Ca maybe.*
Done

*Pg. 2 Line 12: 'Here we' instead of 'We here'*
Done

*Pg. 2 Line 21 move 'and fossil' to the end of the line (so it reads 'below the ship, and fossil foraminifera from sediments')*
Done

*Pg. 2 line 32 – 34: Why poison d18O samples? Does the addition of poison impact cavity-ring down / infrared spectroscopy, as there is some suggestion in the literature that salinity may have an impact on the vaporiser unit (especially changing salinity between samples but also certain salt solutions)*
Answer: The water samples were poisoned during the cruise due to the fact that further parameters will be measured on the same samples (e.g. $\delta^{13}C_{DIC}$). We agree, it is not necessary for $\delta^{18}O$ measurements.

*Pg. 2 line 29/Pg. 3 line 1: Bradshaw (1957) did this massive plankton net study/database of the Pacific, halfway through though he stopped using Rose Ben- gal to identify 'living' as cytoplasm in the shell was just as efficient. Out of curiosity, is there a reason why (Pg. 3 Line 1) 'cytoplasm-bearing' was picked rather than stained? Is there any potential analytical error associated with staining?*
Answer: We had stained and unstained samples. However, to identify living planktic foraminifers it is not necessary to stain the sample with Rose Bengal. The cytoplasm itself is very well visible, which indicates that the foraminifer was still alive during sampling (often greenish-yellowish colour). There is no potential error between stained and unstained samples, however, staining impedes the recognition of reddish tests (e.g. *G. ruber* pink, *G. rubescens*; see Jentzen et al., 2018).

*Pg. 2 line 40: perhaps add 'trilobus-like' to better describe forms of T. sacculifer with a 'spherical last chamber'.*
We added this information.

*Pg. 2 line 41: Globorotalia ungulata, the arch that distinguishes it from G. menardii did you perchance consider laser ablating the arch and the rest of the chamber separately? More out of curiosity, it looks like a chamber feature rather than an embellishment (like a keel) so I wonder is it identical to the rest of the chamber but thicker? Or does it have some structural modification that can be obsered in the ablation profile.*
Answer: We thank the referee for this very interesting comment. Unfortunately, we did not analyse the arch/rim of *G. ungulata* itself.

*Pg. 2 line 41: Globorotalia truncatulinoides dextral, have you seen the new paper by Reynolds et al (2018; Mar. Micro.)? Did you distinguish between crusted and encrusted G. truncatulinoides? Is there any ablation data that might shed light on the Mg/Ca content of encrusted and non-encrusted from your data?*
Answer: We did distinguish between crusted and encrusted specimens. However, we did not measure encrusted *G. truncatulinoides* from the plankton tows, but we measured encrusted specimens of *G. tumida*, which had very low Mg/Ca values (see P9 L40).

*Pg. 3 Lines 1-4. Add in size fraction used for analysis (from the table it appears not all size fractions picked on pg. 2 line 37 were included in the d18O analysis).*
We added this information.

*Pg. 3 Line 14. The magazine/turret of a Kiel device is quite small, perhaps give an indicator of the number of standards per run or in total measured? (as per the Mg/Ca analysis i.e. Pg. 4 Line 22)*
We added this information.

*Pg. 3 Line 18. Add Picarro before model (I assume it's a Picarro instrument based upon L1102-i).*
Yes, we added this information.

*Pg. 3 Line 26. Whose rearrangement of Kim and O'Neil?*
We added this information.

*Pg. 3 Line 31. Add in a mean number of specimens with a plus/minus.*
Done

*Pg. 4 Line 2. The steps were repeated several (1-2) times to completely remove the cytoplasm, could varying this step have any impact on the resultant values between samples that have had a single step and a replicate step?*
Answer: We cannot see any impact on our data by repeating the oxidation step. However, if the samples would not be clean enough, we would expect too high Mg/Ca values caused by the large amount of cytoplasm/organic material from plankton samples.

*Pg. 4 Line 21. Outliers are classified as those values that fall 2 stdev above or below the mean, but that would mean based upon the proportion without (assuming normality) that ∼4.55% of the data could be removed. What's the rationale for this? How much does this influence the data?*
Answer: It does not influence the mean values of our laser ablation measurements. However, this method removes the outliers for the LA-ICP-MS-profiles.

*Pg. 6 Line 18. The transition from juvenile to neanic to adult stages occurs between 100 and 200 um (Brummer et al., 1986 Mar Micro; 1987 Nature), therefore foraminifera above >200 um (species dependent) should be considered as 'adult'. Could the vital effect, described here, instead be due to growth rate (adult specimens with different sized chambers could suggest variation in growth).*
Answer: We thank the referee for this comment. We agree specimens with a test size >200 μm should be noted as adult specimens. However, we think that ontogeny (different stages of growth) influences the isotopic fractionation, which is caused by different metabolism, but also by different numbers of symbionts (e.g. we can expect higher numbers of active symbionts for larger specimens).

*Pg. 7 Line 1. Regarding GAM, if you assume that a shell is a weighted average, if 25% of the shell is composed of gam-calcite and it has a 0.5 per mil offset from living, if you consider it proportionally the gam-calcite would need to have a significantly larger offset from the rest of the living shell.*
Answer: Unfortunately, we cannot give any information of the $\delta^{18}O$ values of the GAM calcite, based on our dataset (unknown $\delta^{18}O$-GAM and unknown % of GAM/individual). GAM calcite of 25% is an estimation for *Orbulina universa*, however, not all individuals/species in the surface sediment have GAM calcite, or up to 25%.

*Pg. 7 Line 34. Would it not be appropriate to (like the palaeotemperature d18O eq1) compare the expected Mg/Ca value by calculating an expected Mg/Ca value (assuming that Mg and Ca concentrations are not limiting and using the values in S2 Table 2)*
Answer: We thank the referee for this comment. However, we think it makes more sense to show the estimated Mg/Ca-temperature (which is species-specific). The ambient seawater temperature is the main environmental factor, which influences the Mg/Ca in foraminiferal tests, and not the concentration of Mg or Ca in the water (if we assume the concentrations are not limited in our case). Based on that we can note

the offset between the estimated Mg/Ca temperature and the seawater temperature (as we can see the offset between $\delta^{18}O_{calcite}$ and $\delta^{18}O_{equilibrium}$, which is T-depending).

*Figure 1. It's a beautiful figure. However, two things, (1.) the map has a transparency that the scale bar does not, could make it difficult to interpret the color, (2.) The rain- bow is a bit difficult for colour blind readers to see (I am guilty of this as well). See: https://www.climate-lab-book.ac.uk/2014/end-of-the-rainbow/*

Answer: We thank the referee for this important comment. 1. We changed the scale bar to transparent. 2. We are aware of the problem for colour-blind readers. We produced the map with ODV program, which gives the rainbow colour automatically, and we regret that we cannot change it to other colours.

*Figure 3. Why not plot the figure like Figure 2 (with isotope, salinity and equilibrium plotted per station) rather than as a composite plot?*

Answer: We plot all stations together because we use the data combined in Fig. 6/8/9

[revised manuscript text omitted]

---

## Author Comment (AC3) · 23 Sep 2018

**Supplement S1**

**S1** Table 1: Stable isotope values ($\delta^{18}O_{calcite}$ and $\delta^{13}C_{calcite}$) of foraminiferal calcite from plankton tows and surface sediments. 1 indicates $\delta^{18}O_{calcite}$ from Steph et al. (2009); # indicates stations of cruise SO164.

| Species | Station | Sampling interval (m) | Size-fraction (µm) | Number of specimens/ sample | $\delta^{18}O_{calcite}$ (‰ VPDB) | $\delta^{13}C_{calcite}$ (‰ VPDB) |
|---|---|---|---|---|---|---|
| *G. sacculifer* | 211-6 | 0–60 | >500 | 3 | -1.56 | 1.20 |
| | 211-6 | | | | -1.49 | 1.24 |
| | 211-6 | | | | -1.44 | 1.24 |
| | 211-6 | | | | -1.50 | 1.51 |
| | 211-5 | | | | -1.41 | 0.91 |
| | 211-5 | | | | -1.49 | 1.68 |
| | 211-5 | | | | -1.51 | 0.85 |
| | 211-5 | | | | -1.61 | 1.02 |
| | 211-6 | | | | -1.45 | 1.23 |
| | 211-5 | | 400–500 | 7 | -1.34 | 0.54 |
| | 211-6 | | 300–400 | 6 | -1.59 | -0.3 |
| | 211-6 | 60–100 | >500 | 3 | -1.06 | 1.26 |
| | 211-5 | | | | -1.40 | 1.35 |
| | 211-6 | | | | -1.39 | 1.21 |
| | 211-5 | | | | -1.30 | 0.70 |
| | 211-5 | | 400–500 | 3 | -1.34 | 0.65 |
| | 211-5 | 100–200 | >500 | 3 | -1.26 | 1.25 |
| | 212-1 | Sediment | 355–400 | 30 | -1.02 | |
| | 212-1 | | | | -1.37 | |
| | 212-1 | | | | -0.95 | |
| | 212-1 | | | | -1.43 | |
| | 219-7 | 0–60 | >500 | 3 | -1.71 | 0.91 |
| | 219-7 | | | | -1.68 | 0.03 |
| | 219-7 | | | | -1.60 | 0.52 |
| | 219-7 | | | | -1.71 | 0.84 |
| | 219-7 | | | | -1.73 | -0.04 |
| | 219-7 | | 400–500 | 5 | -1.34 | -0.05 |
| | 219-8 | | | 6 | -1.79 | -0.43 |
| | 219-8 | | | 6 | -1.76 | 0.21 |
| | 219-8 | | | 5 | -1.77 | -0.08 |
| | 219-7 | | | 5 | -1.69 | 0.45 |
| | 219-8 | | | 5 | -1.55 | 0.07 |
| | 219-7 | | 300–400 | 10 | -1.56 | -0.19 |
| | 219-7 | | | | -1.55 | -0.19 |
| | 219-8 | | | | -1.71 | -0.04 |
| | 219-8 | | | | -1.89 | -0.18 |
| | 219-7 | 60–125 | >500 | 3 | -1.76 | 0.77 |
| | 219-7 | | | | -1.67 | 1.14 |
| | 219-8 | | | | -1.37 | 1.06 |
| | 219-7 | | | | -1.71 | 0.87 |
| | 219-7 | | | | -1.69 | 0.71 |

**Supplement S1**

**S1** Table 1: Continued.

| Species | Station | Sampling interval (m) | Size-fraction (µm) | Number of specimens/ sample | $\delta^{18}O_{calcite}$ (‰ VPDB) | $\delta^{13}C_{calcite}$ (‰ VPDB) |
|---|---|---|---|---|---|---|
| *G. sacculifer* | 219-7 | 60–125 | 400–500 | 5 | -1.39 | 0.41 |
| | 219-7 | | 300–400 | 10 | -1.59 | -0.41 |
| | 219-7 | 125–180 | >500 | 3 | -1.55 | 0.82 |
| | 02-3# | Sediment¹ | 355–400 | | -1.39 | |
| | 220-8 | 0–70 | >500 | 3 | -1.65 | 1.08 |
| | 220-9 | | | 5 | -1.88 | 0.99 |
| | 220-9 | | | 5 | -1.97 | 0.51 |
| | 220-9 | | | 5 | -1.92 | 1.12 |
| | 220-9 | | | 5 | -1.93 | 0.74 |
| | 220-8 | | | 5 | -1.88 | 0.84 |
| | 220-8 | | | 5 | -1.79 | 0.57 |
| | 220-8 | | | 5 | -1.65 | 1.35 |
| | 220-9 | | 400–500 | 9 | -1.76 | 0.08 |
| | 220-9 | | | 9 | -1.56 | 0.96 |
| | 220-9 | | | 12 | -1.73 | 0.34 |
| | 220-8 | 70–100 | >500 | 3 | -1.65 | 1.52 |
| | 220-8 | | 400–500 | 5 | -1.47 | 0.70 |
| | 22-2# | Sediment¹ | 355–400 | | -1.25 | |
| | 221-8 | 0–40 | >500 | 3 | -1.79 | 0.83 |
| | 221-8 | | 400–500 | 5 | -1.83 | 1.73 |
| | 221-7 | | | 7 | -2.07 | 0.39 |
| | 221-8 | | | 8 | -1.99 | 0.82 |
| | 221-8 | | | 7 | -1.96 | 0.96 |
| | 221-8 | | 300–400 | 9 | -1.94 | 0.18 |
| | 221-8 | | | 9 | -1.99 | 0.35 |
| | 221-8 | | | 9 | -1.98 | 0.41 |
| | 221-8 | | | 9 | -2.03 | 0.46 |
| | 221-8 | | | 7 | -2.08 | 1.00 |
| | 221-8 | | | 7 | -1.96 | 0.68 |
| | 221-7 | 40–60 | 400–500 | 5 | -1.66 | 1.26 |
| | 221-8 | | 300–400 | 6 | -1.66 | 0.98 |
| | 221-8 | 60–150 | 400–500 | 5 | -1.59 | 1.67 |
| | 24-3# | Sediment¹ | 355–400 | | -1.5 | |
| | 222-7 | 0–40 | >500 | 3 | -1.52 | 2.13 |
| | 222-6 | | 400–500 | 5 | -1.29 | 0.73 |
| | 222-7 | | 300–400 | 7 | -1.94 | 1.42 |
| | 222-6 | 40–80 | 300–400 | 5 | -1.68 | 1.33 |
| | 222-8 | Sediment | 355–400 | 30 | -1.59 | |
| *O. universa* | 211-6 | 0–60 | >500 | 10 | -1.33 | 2.38 |
| | 211-5 | | | 5 | -1.58 | 1.83 |
| | 211-5 | | | 5 | -1.54 | 2.21 |
| | 211-6 | | | 8 | -1.24 | 1.35 |
| | 211-5 | 60–100 | >500 | 5 | -1.23 | 1.22 |
| | 211-5 | | | 5 | -1.20 | 1.16 |
| | 211-6 | | | 5 | -1.39 | 1.30 |
| | 211-6 | | | 10 | -1.24 | 1.32 |
| | 211-6 | 100–200 | >500 | 7 | -0.85 | 0.99 |

**Supplement S1**

**S1** Table 1: Continued.

| Species | Station | Sampling interval (m) | Size-fraction (μm) | Number of specimens/ sample | δ¹⁸O_calcite (‰ VPDB) | δ¹³C_calcite (‰ VPDB) |
|---|---|---|---|---|---|---|
| *O. universa* | 212-1 | Sediment | 355–400 | 10 | -0.73 | |
| | 220-9 | 0–70 | >500 | 6 | -1.55 | 1.84 |
| | 22-2# | Sediment | 355–400 | 18 | -1.33 | |
| | 22-2# | | | | -1.14 | |
| | 221-8 | 0–40 | >500 | 10 | -1.82 | 1.63 |
| | 221-7 | | | 9 | -1.84 | 1.39 |
| | 221-8 | 40–60 | >500 | 7 | -1.6 | 1.47 |
| | 221-7 | 60–150 | >500 | 5 | -1.42 | 1.25 |
| | 24-3# | Sediment | 355–400 | 18 | -1.21 | |
| | 24-3# | | | | -1.88 | |
| | 24-3# | | | | -1.40 | |
| *N. dutertrei* | 221-8 | 0–40 | 400–500 | 3 | -1.28 | 2.65 |
| | 221-8 | | 300–400 | 5 | -1.68 | 2.02 |
| | 221-7 | | 250–300 | 6 | -1.65 | 1.61 |
| | 221-7 | | | 5 | -1.81 | 0.88 |
| | 221-8 | | | 6 | -1.25 | 1.67 |
| | 221-8 | | | 6 | -1.77 | 1.42 |
| | 221-8 | 40–60 | 400–500 | 2 | -1.35 | 2.12 |
| | 221-7 | | 300–400 | 3 | -1.58 | 2.2 |
| | 221-7 | | 250–300 | 6 | -2.12 | 1.05 |
| | 221-8 | 60–150 | 400–500 | 2 | -1.28 | 1.59 |
| | 221-8 | | 300–400 | 3 | -1.39 | 1.98 |
| | 221-7 | | 250–300 | 6 | -1.44 | 0.53 |
| | 24-3# | Sediment[1] | 355–400 | | -0.53 | |
| | 222-7 | 0–40 | 300–400 | 5 | -1.27 | 1.03 |
| | 222-8 | Sediment | 355–400 | 13 | -0.26 | |
| *P. obliquiloculata* | 211-5 | 0–60 | 300–400 | 3 | -0.83 | 0.05 |
| | 212-1 | Sediment | 355–400 | 18 | -0.14 | |
| | 212-1 | | | | -0.11 | |
| | 212-1 | | | | -0.16 | |
| | 219-8 | 60–125 | 300–400 | 3 | -1.15 | 0.06 |
| | 02-3# | Sediment | 355–400 | 12 | -0.87 | |
| | 220-8 | 0–70 | 300–400 | 3 | -1.24 | 0.12 |
| | 220-8 | 110–150 | 300–400 | 3 | -0.98 | 0.16 |
| | 22-2# | Sediment | 355–400 | 12 | -0.65 | |
| | 22-2# | | | | -0.91 | |
| | 22-2# | | | | -0.67 | |
| | 221-7 | 0–40 | 300–400 | 3 | -1.48 | 0.03 |
| | 221-8 | | | | -0.23 | 0.51 |
| | 221-8 | | | | -1.47 | 0.24 |
| | 221-8 | 40–60 | 300–400 | 3 | -1.41 | -0.07 |
| | 221-8 | 60–150 | 300–400 | 3 | -1.01 | 0.09 |
| | 221-7 | | | | -0.74 | 0.29 |
| | 221-7 | | | | -1.68 | -0.20 |
| | 24-3# | Sediment | 355–400 | 11 | -0.99 | |
| | 24-3# | | | | 0.05 | |

**Supplement S1**

**S1** Table 1: Continued.

| Species | Station | Sampling interval (m) | Size-fraction (μm) | Number of specimens/ sample | $\delta^{18}O_{calcite}$ (‰ VPDB) | $\delta^{13}C_{calcite}$ (‰ VPDB) |
|---|---|---|---|---|---|---|
| *P. obliquiloculata* | 222-6 | 0–40 | 300–400 | 3 | -1.22 | 0.22 |
| *G. menardii* | 211-6 | 60–100 | 300–400 | 3 | -1.01 | 0.78 |
| | 211-6 | 100–200 | 300–400 | 5 | -0.16 | 0.21 |
| | 212-1 | Sediment | 355–400 | 8 | -0.46 | |
| | 212-1 | | | | -0.85 | |
| | 212-1 | | | | -1.25 | |
| | 221-8 | 60–150 | 300–400 | 5 | -1.17 | 0.59 |
| | 221-8 | 150–210 | 400–500 | 2 | -0.87 | 0.49 |
| | 24-3# | Sediment[1] | 355–400 | | -0.24 | |
| *G. ungulata* | 211-5 | 0–60 | 400–500 | 4 | -1.08 | 0.99 |
| | 211-6 | | | | -0.98 | 1.33 |
| | 211-6 | | | | -0.67 | 0.70 |
| | 211-5 | | | | -0.70 | 0.99 |
| | 211-6 | | 300–400 | 5 | -1.03 | 1.08 |
| | 211-5 | | | 4 | -1.14 | 1.24 |
| | 211-5 | | | 4 | -0.71 | 0.64 |
| | 211-6 | | | 4 | -0.92 | 1.04 |
| | 211-5 | | 250–300 | 7 | -0.99 | 1.10 |
| | 211-5 | | | | -1.09 | 1.21 |
| | 211-6 | | | | -1.06 | 1.34 |
| | 211-6 | | | | -0.90 | 1.27 |
| | 211-6 | | | | -0.99 | 1.09 |
| | 211-6 | 60–100 | 300–400 | 5 | -0.90 | 0.65 |
| | 211-5 | | | 5 | -0.88 | 0.44 |
| | 211-6 | | | 6 | -1.07 | 0.93 |
| | 211-6 | 100–200 | 400–500 | 3 | -0.27 | 0.63 |
| | 211-5 | | 300–400 | 5 | -0.20 | 0.46 |
| | 211-5 | | | 5 | -0.42 | 0.37 |
| | 211-6 | | | 4 | -0.14 | 0.52 |
| | 212-1 | Sediment | 355–400 | 12 | 0.30 | |
| | 212-1 | | | | -1.20 | |
| | 212-1 | | | | -0.95 | |
| | 212-1 | | | | -0.83 | |
| *G. truncatulinoides* d. | 211-6 | 100–200 | 300–400 | 4 | -0.10 | -0.44 |
| | 211-6 | 200–300 | 300–400 | 2 | 0.89 | -0.02 |
| | 212-1 | Sediment | 355–400 | 7 | 1.35 | |
| | 212-1 | | | | 1.29 | |
| | 212-1 | | | | 0.95 | |
| | 219-7 | 220–400 | 300–400 | 2 | 0.18 | 0.10 |
| | 02-3# | Sediment[1] | 355–400 | | 0.98 | |
| | 220-8 | 150–220 | 300–400 | 2 | 0.20 | 0.14 |
| | 22-2# | Sediment[1] | 355-400 | | 0.8 | |
| | 221-7 | 150–210 | 300–400 | 4 | 0.01 | -0.08 |
| | 221-7 | 210–300 | 300–400 | 3 | 0.52 | 0.05 |
| | 24-3# | Sediment[1] | 355–400 | | 1.54 | |
| *G. tumida* | 219-8 | 180–220 | 400–500 | 2 | -0.88 | 0.87 |
| | 219-8 | 220–400 | 400–500 | 3 | -0.28 | 0.84 |
| | 02-3# | Sediment[1] | 355–400 | | -0.11 | |

**Supplement S1**

**S1** Table 2**:** Mg/Ca ratios of foraminiferal calcite measured on bulk foraminiferal samples (ICP-OES) and on single chambers (LA-ICP-MS) from plankton tows and surface sediments. For the LA-ICP-MS measurements, average values (± standard deviation) of all chambers from single specimens are calculated. 2 indicates Mg/Ca ratios from Regenberg et al. (2006), *with dissolution correction (published in Regenberg et al., 2009); # stations of cruise SO164.

| Species | Station | Sampling interval (m) | Size fraction (μm) | Number ind. /sample | Mg/Ca (mmol mol⁻¹) ICP-OES | Chambers /ind. | Mg/Ca (mmol mol⁻¹) LA-ICP-MS |
|---|---|---|---|---|---|---|---|
| *G. sacculifer* | 211-6 | 0–60 | >500 | 12 | 3.95 | | |
| | 211-6 | | | 12 | 3.08 | | |
| | 211-6 | | | 12 | 3.49 | | |
| | 211-6 | | | 12 | 3.26 | | |
| | 211-6 | | | 12 | 3.35 | | |
| | 211-5 | | | 12 | 3.64 | F, F-1, F-2 /tri83 | 3.27 ± 0.24 |
| | 211-5 | | | 12 | 3.46 | | |
| | 211-5 | | | 12 | 3.37 | | |
| | 211-5 | | | 12 | 3.26 | | |
| | 211-5 | | | 12 | 3.53 | | |
| | 211-5 | | | 12 | 3.37 | | |
| | 211-6 | | | 12 | 3.6 | | |
| | 211-6 | | | 12 | 3.44 | | |
| | 211-6 | | 400–500 | 42 | 3.51 | | |
| | 211-5/211-6 | | 300–400 | 64 | 3.72 | | |
| | 211-5 | 60–100 | >500 | 12 | 3.44 | F, F-1, F-2 /tri85 | 3.95 ± 0.8 |
| | 211-6 | | | 12 | 3.63 | | |
| | 211-6 | | 400–500 | 10 | 3.94 | | |
| | 211-5 | 100–200 | >500 | - | - | F, F-1, F-2 /tri87 | 2.48 ± 0.57 |
| | 212-1 | Sediment | 355–400 | 30 | 4.44 | | |
| | 212-1 | | | | 4.17 | | |
| | 212-1 | | | | 4.39 | | |
| | 212-1 | | | | 4.39 | | |
| | 219-7 | 0–60 | >500 | 16 | 3.9 | | |
| | 219-8 | | | | 4.24 | F, F-1, F-2 /tri6 | 3.11 ± 0.7 |
| | 219-8 | | | | 4.16 | F, F-1, F-2 /tri7 | 3.87 ± 0.2 |
| | 219-7 | | 400–500 | 40 | 4.37 | | |
| | 219-7/219-8 | | 300–400 | 70 | 4.29 | | |
| | 219-8 | 60–125 | >500 | 17 | 4.30 | F, F-1, F-2 /tri8 | 3.18 ± 0.62 |
| | 219-7/219-8 | | 400–500 | 24 | 4.22 | | |
| | 219-7/219-8 | | 300–400 | 54 | 4.37 | | |
| | 219-8 | 220–400 | >500 | - | - | F, F-1, F-2 /tri12 | 3.86 ± 0.7 |
| | 02-3# | Sediment² | 355–400 | | 4.2 | | |
| | 220-9 | 0–70 | >500 | 10 | 3.87 | F, F-1, F-2 /tri2 | 3.72 ± 0.44 |
| | 220-8 | | | 11 | 4.15 | F, F-1,F-2 /tri99 | 2.82 ± 0.56 |
| | 220-9 | | | 10 | 4.15 | | |
| | 220-8 | | | 12 | 4.2 | | |
| | 220-8 | | | 12 | 4.14 | | |
| | 220-9 | | | 11 | 4.36 | | |
| | 220-9 | | | 12 | 4.08 | | |
| | 220-8 | | | 11 | 4.0 | | |
| | 220-8 | | 400–500 | 38 | 4.20 | | |
| | 220-8 | | 300–400 | 80 | 3.90 | | |
| | 220-8 | 70–110 | >500 | - | - | F, F-1, F-2 /tri100 | 3.12 ± 1.00 |
| | 220-9 | 150–220 | 250–300 | - | - | F, F-1 /tri4 | 3.7 ± 0.02 |
| | 22-2# | Sediment² | 355–400 | | 3.71/4.45* | | |

**S1** Table 2**:** Continued.

| Species | Station | Sampling interval (m) | Size fraction (µm) | Number ind. /sample | Mg/Ca (mmol mol⁻¹) ICP-OES | Chambers /ind. | Mg/Ca (mmol mol⁻¹) LA-ICP-MS |
|---|---|---|---|---|---|---|---|
| *G. sacculifer* | 221-8 | 0–40 | >500 | 20 | 4.00 | | |
| | 221-7 | | | | | F,F1, F2 /tri101 | 4.3 ± 0.25 |
| | 221-7 | | 400–500 | 80 | 4.01 | | |
| | 221-7 | | 300–400 | 86 | 3.91 | | |
| | 221-7 | 40–60 | >500 | - | - | F, F1, F2 /tri19 | 3.3 ± 0.53 |
| | 221-8 | | 400–500 | 15 | 4.02 | - | |
| | 221-7/221-8 | | 300–400 | 54 | 4.06 | - | |
| | 221-7 | 60–150 | >500 | - | - | F, F1, F2 /tri20 | 3.8 ± 0.35 |
| | 24-3# | Sediment² | 355–400 | | 4.23 | | |
| | 222-6 | 0–40 | >500 | - | - | F, F-1, F-2 /tri110 | 4.0 ± 0.14 |
| | 222-7 | | 400–500 | 12 | 4.55 | | |
| | 222-6 | | 300–400 | 30 | 4.13 | | |
| | 222-7 | 40–80 | 400–500 | - | - | F, F-1, F-2 /tri111 | 4.0 ± 0.3 |
| | 222-7 | 80–120 | >500 | - | - | F, F-1, F-2 /tri23 | 3.27 ± 0.65 |
| | 222-8 | Sediment | 355–400 | 30 | 3.84 | | |
| *N. dutertrei* | PF 12 | 3.5 | 365 | - | - | F, F-1, F-2 /dut116 | 2.7 ± 0.74 |
| | 02-3# | Sediment² | 355–400 | | 2.58/2.86* | | |
| | 22-2# | Sediment² | 355–400 | | 1.84/3.15* | | |
| | 221-7 | 0–40 | 300–400 | - | - | F, F-1, F-2 /dut104 | 1.73 ± 0.38 |
| | 221-7/221-8 | | | 50 | 2.97 | | |
| | 221-7/221-8 | 40–60 | | 19 | 4.21 | | |
| | 24-3# | Sediment² | 355–400 | | 2.63 | | |
| | 222-6 | 0–40 | 300–400 | - | - | F, F-1, F-2 /dut112 | 2.99 ± 0.77 |
| *G. ungulata* | PF 7 | 3.5 | 425 | - | - | F, F-1, F-2 /ung113 | 2.35 ± 0.35 |
| | PF 12 | | 450 | - | - | F, F-1, F-2 /ung117 | 2.55 ± 0.15 |
| | 211-5 | 0–60 | >500 | - | - | F-1, F-2 /ung28 | 3.19 ± 0.32 |
| | 211-5/221-6 | | 400–500 | 14 | 3.39 | | |
| | 211-5/221-6 | | 300–400 | 28 | 3.32 | | |
| | 211-5 | 60–100 | >500 | - | - | F, F-1, F-2 /ung29 | 3.22 ± 0.41 |
| | 211-5/211-6 | | 400–500 | 14 | 3.19 | | |
| | 211-5/211-6 | | 300–400 | 22 | 3.33 | | |
| | 211-5 | 100–200 | >500 | - | - | F, F-1, F-2 /ung30 | 3.1 ± 0.06 |
| | 211-5/211-6 | | 400–500 | 17 | 3.48 | | |
| | 211-5/211-6 | | 300–400 | 20 | 3.17 | | |
| *O. universa* | 211-6 | 0–60 | >500 | - | - | F /uni36 | 10.3 ± 0.33 |
| | 211-6 | 60–100 | >500 | - | - | F /uni37 | 10.09 ± 0.54 |
| | 219-8 | 0–60 | >500 | - | - | F /uni39 | 9.08 ± 0.95 |
| | 219-7 | 60–125 | >500 | - | - | F /uni40 | 7.13 ± 0.68 |
| | 219-7 | 180–220 | >500 | - | - | F /uni41 | 8.31 ± 1.5 |
| | 220-8 | 0–70 | >500 | - | - | F /uni42 | 8.16 ± 0.50 |
| | 220-8 | 110–150 | >500 | - | - | F /uni44 | 7.05 ± 0.18 |
| | 221-8 | 0–40 | >500 | - | - | F /uni106 | 7.28 ± 0.06 |
| | 221-8 | 40–60 | >500 | - | - | F /uni46 | 8.09 ± 0.4 |
| | 221-8 | 60–150 | >500 | - | - | F /uni47 | 9.79 ± 0.4 |
| | 222-7 | 0–40 | >500 | - | - | F /uni48 | 6.55 ± 0.16 |
| | 222-7 | 120–180 | >500 | - | - | F /uni50 | 5.3 ± 0.08 |

**S1** Table 2**:** Continued.

| Species | Station | Sampling interval (m) | Size fraction (μm) | Number ind. /sample | Mg/Ca (mmol mol⁻¹) ICP-OES | Chambers /ind. | Mg/Ca (mmol mol⁻¹) LA-ICP-MS |
|---|---|---|---|---|---|---|---|
| *G. menardii* | PF 19 | 3.5 | 355 | - | - | F, F-1, F-2 /men118 | 1.76 ± 0.31 |
| | 211-5 | 100–200 | 300–400 | - | - | F, F-1, F-2 /men91 | 3.45 ± 0.27 |
| | 211-5 | | 300–400 | - | - | F, F-1, F-2 /men92 | 2.62 ± 0.95 |
| | 211-5 | 200–300 | >500 | - | - | F, F-1, F-2 /men31 | 2.8 ± 0.5 |
| | 02-3# | Sediment² | 355–400 | | 3.49/3.52* | | |
| | 220-8 | 0–70 | 400–500 | - | - | F, F-1, F-2 /men26 | 2.21 ± 0.17 |
| | 22-2# | Sediment² | 355–400 | | 2.20/3.31* | | |
| | 221-7 | 0–40 | 400–500 | - | - | F, F-1, F-2 /men32 | 3.18 ± 0.19 |
| | 221-8 | 60–150 | 400–500 | - | - | F, F-1, F-2 /men105 | 3.36 ± 0.52 |
| | 221-7 | 150–210 | >500 | - | - | F, F-1, F-2 /men34 | 3.24 ± 0.42 |
| | 221-8 | 210–300 | >500 | - | - | F, F-1, F-2 /men35 | 3.69 ± 0.49 |
| | 24-3# | Sediment² | 400–500 | | 2.98 | | |
| | 222-6 | 0-40 | 400–500 | - | - | F, F-1, F-2 /men27 | 3.92 ± 0.31 |
| *G. truncatulinoides* d. | PF 11 | 3.5 | 325 | - | - | F, F-1, F-2, F-3 / tdex115 | 3.22 ± 1.56 |
| | 211-5 | 100–200 | 300–400 | - | - | F-1, F-2, F-3 / tdex90 | 3.0 ± 0.55 |
| | 22-2# | Sediment² | 355–400 | | 1.62/2.76* | | |
| | 221-8 | 150–210 | 300–400 | - | - | F, F-1, F-2, F-3 / tdex108 | 1.66 ± 0.19 |
| | 221-8 | 210–300 | 400–500 | - | - | F, F-1, F-2, F-3 / tdex109 | 2.84 ± 0.53 |
| | 24-3# | Sediment² | 400–500 | | 2.28 | | |
| *G. tumida* | 219-8 | 60–125 | >500 | - | - | F, F-1, F-2 /tum61 | 2.45 ± 0.13 |
| | 219-8 | 125–180 | >500 | - | - | F ,F-1, F-2 /tum62 | 1.6 ± 0.38 |
| | 219-7 | 180–220 | >500 | - | - | F, F-1, F-2 /tum63 | 2.25 ± 0.35 |
| | 219-7 | 220–400 | >500 | - | - | F, F-1, F-2 /tum64 | 1.57 ± 0.62 |
| | 02-3# | Sediment² | 400–500 | | 2.43 | | |
| | 22-2# | Sediment² | 355–400 | | 1.95/2.93* | | |
| *P. obliquiloculata* | 221-8 | 0–40 | 400–500 | - | - | F, F-1 /obli77 | 2.54 ± 0.09 |
| | 221-7 | 40–60 | 400–500 | - | - | F, F-1 /obli78 | 2.55 ± 0.31 |
| | 222-7 | 0–40 | 300–400 | - | - | F, F-1, F-2 /obli80 | 3.44 ± 1.01 |
| | 222-7 | 40–80 | 300–400 | - | - | F, F-1, F-2 /obli81 | 2.43 ± 0.51 |
| | 222-6 | 80–120 | 300–400 | - | - | F ,F-1, F-2 /obli82 | 3.3 ± 0.32 |

**Supplement S1**

**S1** Table 3**:** Stable isotope values in seawater ($\delta^{18}O_{seawater}$), measured temperature (°C) and salinity (psu) during RV *Meteor* cruise M78/1 (Schönfeld et al., 2011, by courtesy of C. Dullo and S. Flögel).

| Station | Sampling depth (m) | $\delta^{18}O_{seawater}$ (‰ VSMOW) | Temperature (°C) | Salinity (psu) |
|---------|--------------------|--------------------------------------|------------------|----------------|
| 210-13 | 40 | 0.98 | 24.8 | 36.0 |
| 210-13 | 85 | 1.02 | 24.2 | 36.2 |
| 210-13 | 100 | 1.01 | 24.0 | 36.8 |
| 210-13 | 150 | 1.05 | 21.0 | 36.4 |
| 210-13 | 190 | 0.92 | 19.2 | 36.4 |
| 210-13 | 275 | 0.75 | 15.9 | 36.1 |
| 210-13 | 400 | 0.45 | 11.8 | 35.4 |
| 219-1 | 50 | 0.96 | 26.1 | 35.9 |
| 219-1 | 100 | 0.94 | 26.1 | 36.0 |
| 219-1 | 220 | 0.96 | 19.5 | 36.6 |
| 219-1 | 600 | 0.27 | 8.5 | 34.9 |
| 220-1 | 10 | 0.97 | 26.2 | 35.7 |
| 220-1 | 61 | 1.02 | 26.1 | 35.7 |
| 220-1 | 91 | 1.21 | 26.1 | 36.8 |
| 220-2 | 136 | 1.17 | 22.1 | 36.8 |
| 220-2 | 196 | 1.04 | 18.4 | 36.5 |
| 220-2 | 485 | 0.3 | 9.3 | 35.0 |
| 221-1 | 10 | 0.97 | 26.4 | 35.5 |
| 221-1 | 30 | 1.01 | 26.4 | 35.5 |
| 221-1 | 60 | 1.21 | 26.5 | 36.6 |
| 221-2 | 100 | 1.28 | 24.0 | 37.2 |
| 221-2 | 150 | 1.11 | 20.2 | 36.8 |
| 221-2 | 200 | 0.99 | 17.7 | 36.4 |
| 221-2 | 500 | 0.31 | 8.9 | 34.9 |
| 222-1 | 10 | 1.0 | 26.5 | 35.7 |
| 222-1 | 30 | 1.0 | 26.6 | 35.7 |
| 222-1 | 55 | 1.12 | 22.7 | 36.7 |
| 222-1 | 75 | 1.11 | 21.8 | 36.8 |
| 222-1 | 140 | 1.04 | 18.3 | 36.5 |
| 222-1 | 229 | 0.74 | 14.4 | 35.7 |

**Supplement S1**

**S1** Table 4**:** Data (average values) of the Thermosalinograph during cruises M78/1 (Schönfeld et al., 2009).

| Cruise | Station Nr. | Temperature (°C) | Salinity (psu) |
|--------|-------------|------------------|----------------|
| M78/1 | 1 | 26.65 | 35.93 |
| M78/1 | 2 | 25.60 | 36.00 |
| M78/1 | 3 | 26.87 | 35.54 |
| M78/1 | 4 | 26.60 | 35.80 |
| M78/1 | 5 | 26.00 | 35.70 |
| M78/1 | 6 | 25.50 | 35.70 |
| M78/1 | 7 | 24.94 | 35.93 |
| M78/1 | 10 | 20.00 | 36.30 |
| M78/1 | 11 | 20.00 | 36.40 |
| M78/1 | 14 | 19.90 | 36.40 |
| M78/1 | 15 | 20.00 | 36.40 |
| M78/1 | 16 | 20.10 | 36.40 |
| M78/1 | 17 | 20.00 | 36.40 |
| M78/1 | 18 | 20.00 | 36.50 |
| M78/1 | 19 | 20.50 | 36.40 |
| M78/1 | 20 | 20.00 | 36.40 |
| M78/1 | 21 | 20.20 | 36.40 |
| M78/1 | 23 | 24.40 | 35.90 |
| M78/1 | 24 | 24.20 | 35.90 |
| M78/1 | 33 | 24.40 | 35.70 |
| M78/1 | 34 | 26.00 | 35.20 |
| M78/1 | 35 | 25.90 | 35.20 |
| M78/1 | 36 | 26.16 | 33.46 |
| M78/1 | 37 | 26.30 | 31.10 |
| M78/1 | 39 | 25.30 | 33.50 |
| M78/1 | 40 | 25.90 | 35.30 |
| M78/1 | 41 | 26.70 | 34.60 |
| M78/1 | 42 | 26.40 | 34.80 |
| M78/1 | 43 | 26.70 | 34.90 |
| M78/1 | 44 | 26.90 | 34.90 |
| M78/1 | 45 | 27.00 | 34.50 |
| M78/1 | 47 | 27.20 | 34.60 |

[Figure]

**S2** Figure 1: Assessment of existing δ¹⁸O-paleotemperature relationships. Grey dots: Difference between the measured δ¹⁸O$_{calcite}$ and the measured δ¹⁸O$_{seawater}$, depicted at the average in situ temperature of the plankton net intervals measured during cruise M78/1. Black error bars denote the temperature ranges of the sampling intervals. Coloured-coded lines labelled by numbers are published δ¹⁸O-paleotemperature equations (cf. Supplement S2 Table 1).

**Supplement S2**

**S2** Table 1**:** Temperature: $\delta^{18}O$ relationship from different studies including different conversion factors (SMOW to V-PDB; cf. Bemis et al., 1998). A = species-specific equation used to estimate $\delta^{18}O_{seawater}$ for *G. sacculifer*.

| colspan | | | | | | | |
|---|---|---|---|---|---|---|---|
| $T= \mathbf{a} + \mathbf{b}*(\delta^{18}O_{calcite} - \delta^{18}O_{seawater}) + \mathbf{c}*(\delta^{18}O_{calcite} - \delta^{18}O_{seawater})^2$ | | | | | | | |
| Nr. | Reference | Species | Material | **a** | **b** | **c** | SMOW to V-PDB conversion |
| 1 | Kim and O´Neil 1997 | Inorganic | Experiment | 16.1 | -4.64 | 0.09 | -0.27 |
| 2 | Shackleton 1974 | *Uvigerina* | Sediment | 16.9 | -4.38 | 0.1 | -0.20 |
| 3 | Erez and Luz 1983 | *G. sacculifer* | Culture experiment | 17.0 | -4.52 | 0.03 | -0.22 |
| 4 | Bouvier-Soumagnac and Duplessy 1985 | *O. universa* | Culture experiment | 16.4 | -4.67 | | -0.20 |
| 5 | Bouvier-Soumagnac and Duplessy 1985 | *O. universa* | Plankton tow | 15.4 | -4.81 | | -0.20 |
| 6 | Bouvier-Soumagnac and Duplessy 1985 | *G. menardii* | Plankton tow | 14.6 | -5.03 | | -0.20 |
| 7 | Bouvier-Soumagnac and Duplessy 1985 | *N. dutertrei* | Plankton tow | 10.5 | -6.58 | | -0.20 |
| 8 | Bemis et al. 1998 | *O. universa* | Culture experiment, high-light conditions | 14.9 | -4.8 | | -0.27 |
| 9 | Bemis et al. 1998 | *O. universa* | Culture experiment, low-light conditions | 16.5 | -4.8 | | -0.27 |
| 10 | Mulitza et al. 2003 | *G. sacculifer* | Surface pump samples | 14.91 | -4.35 | | -0.27 |
| 11 | Spero et al. 2003 | *G. sacculifer* | A Culture experiment, high-light conditions | 12.0 | -5.67 | | -0.27 |
| 12 | Farmer et al. 2007 | *G. sacculifer* | Surface sediment | 16.2 | -4.94 | | -0.27 |
| 13 | Farmer et al. 2007 | *O. universa* | Surface sediment | 16.5 | -5.11 | | -0.27 |
| 14 | Farmer et al. 2007 | *N. dutertrei* | Surface sediment | 14.6 | -5.09 | | -0.27 |
| 15 | Farmer et al. 2007 | *G. menardii* | Surface sediment | 16.6 | -5.20 | | -0.27 |
| 16 | Farmer et al. 2007 | *P. obliquiloculata* | Surface sediment | 16.8 | -5.22 | | -0.27 |
| 17 | Farmer et al. 2007 | *G. tumida* | Surface sediment | 13.1 | -4.95 | | -0.27 |

[Figure]

**S2** Figure 2: Average Mg/Ca values (±standard deviations) of LA-ICP-MS measurements of single tests vs. in situ temperature (recorded during M78/1). Brown triangles: Mg/Ca values of living specimens depicted at the average in situ temperature of the plankton net intervals (MSN and PF) during cruise M78/1. Black error bars indicate the standard deviations of single foraminiferal tests (cf. Supplement S1) and temperature ranges of the sampling intervals, respectively. The various published Mg/Ca calibration curves are colour-coded and labelled by numbers (cf. Supplement S2 Table 2).

**Supplement S2**

**S2** Table 2: Relationship between temperature and Mg/Ca ratios from different authors, species and material. A–H indicate species-specific calibrations used to estimate calcification temperature from Mg/Ca for A=*G. sacculifer*; B=*O. universa*; C=*N. dutertrei*; D=*P. obliquiloculata*; E=*G. menardii*; F=*G. ungulata*; G=*G. truncatulinoides* dextral; H=*G. tumida*.

| Mg/Ca = **b** * exp(**a** * T) | | | | | |
|---|---|---|---|---|---|
| Nr. | Reference | Species | | Material/Method | **b** | **a** |
| 1 | Regenberg et al. 2009 | *G. sacculifer* | A | Surface sediment/ICP-OES | 0.596 | 0.075 |
| 2 | Nürnberg et al. 2000 | *G. sacculifer* | | ICP-OES | 0.491 | 0.076 |
| 3 | Regenberg et al. 2009 | *N. dutertrei* | C | Surface sediment/ICP-OES | 0.65 | 0.065 |
| 4 | Regenberg et al. 2009 | *G. tumida* | H | Surface sediment/ICP-OES | 1.23 | 0.041 |
| 5 | Regenberg et al. 2009 | *G. menardii* | E | Surface sediment/ICP-OES | 0.36 | 0.091 |
| 6 | Regenberg et al. 2009 | *G. truncatulinoides* d. | | Surface sediment/ICP-OES | 1.32 | 0.05 |
| 7 | Regenberg et al. 2009 | Shallow-dweller | F | Surface sediment/ICP-OES | 0.29 | 0.101 |
| 8 | Regenberg et al. 2009 | Deep-dweller | | Surface sediment/ICP-OES | 0.84 | 0.083 |
| 9 | Russel et al. 2004 | *O. universa* | B | Culture experiments/ICP-MS | 0.85 | 0.096 |
| 10 | Lea et al. 1999 | *O. universa* | | Culture experiments/ICP-MS | 1.36 | 0.085 |
| 11 | Anand et al. 2003 | *N. dutertrei* | | Sediment-Trap/ICP-OES | 0.342 | 0.09 |
| 12 | Anand et al. 2003 | *G. sacculifer* | | Sediment-Trap/ICP-OES | 1.06 | 0.048 |
| 13 | Anand et al. 2003 | *P. obliquiloculata* | | Sediment-Trap/ICP-OES | 0.18 | 0.12 |
| 14 | Anand et al. 2003 | *P. obliquiloculata* | | Sediment-Trap/ICP-OES | 0.328 | 0.09 |
| 15 | Anand et al. 2003 | *G. truncatulinoides* d. | | Sediment-Trap/ICP-OES | 0.359 | 0.09 |
| 16 | Anand et al. 2003 | *O. universa* | | Sediment-Trap/ICP-OES | 0.595 | 0.09 |
| 17 | Anand et al. 2003 | Multi-species | | Sediment-Trap/ICP-OES | 0.38 | 0.09 |
| 18 | Elderfield and Ganssen 2000 | Multi-species | | Surface sediment/ICP-OES | 0.52 | 0.1 |
| 19 | Nürnberg et al. 1996 | *G. sacculifer* | | Culture experiment/EPMA | 0.39 | 0.09 |
| 20 | Dekens et al. 2002 | *G. sacculifer* | | Surface sediment/ICP-MS | 0.37 | 0.09 |
| 21 | Cléroux et al. 2008 | *G. truncatulinoides* d. | G | Surface sediment/ICP-AES | 0.62 | 0.074 |
| 22 | Cléroux et al. 2008 | *P. obliquiloculata* | D | Surface sediment/ICP-AES | 1.02 | 0.039 |
| 23 | McKenna and Prell 2004 | *G. truncatulinoides* d. | | Surface sediment/EPMA | 0.355 | 0.098 |

**Supplement S3**

**S3** Table 1**:** Spearman rank correlation obtained from PAST (Hammer et al., 2001).

| Species | $\delta^{18}O_{calcite}$ Two tailed probability | $\delta^{18}O_{calcite}$ Correlation value | $\delta^{13}C_{calcite}$ Two tailed probability | $\delta^{13}C_{calcite}$ Correlation value |
|---|---|---|---|---|
| *G. sacculifer* | 0.00 | 0.34 | 0.00 | 0.45 |
| *G. ungulata* | 0.28 | 0.25 | 0.43 | -0.19 |
| *G. menardii* | 0.90 | -0.10 | 0.93 | -0.07 |
| *N. dutertrei* | 0.04 | 0.57 | 0.02 | 0.64 |

**Supplement S4**

[Figure]

**S4** Figure 1**:** Average Mg/Ca values (± standard deviation) of single chambers (F, F-1 and F-2) from 17 specimens of *G. sacculifer*. Single individuals were collected at different stations and water depth intervals (cf. Supplement S1).

[Figure]

**S4** Figure 2: Laser ablation ICP-MS profiles of Mg/Ca (average values) through *O. universa*. Spherical chambers were measured three times from the outside of the tests toward the inside (left to right). Single individuals were collected at different stations and water depth intervals (cf. Supplement S1).

[Figure]

**S4** Figure 3: Laser ablation ICP-MS profiles of Mg/Ca through *P. obliquiloculata*. Single chambers (F, F-1 and F-2) were measured from the outside of the tests toward the inside (left to right). Single individuals were collected at different stations and water depth intervals (cf. Supplement S1).

[Figure]

**S4** Figure 4**:** Laser ablation ICP-MS profiles of Mg/Ca through *N. dutertrei*. Single chambers (F, F-1 and F-2) were measured from the outside of the tests toward the inside (left to right). Single individuals were collected at different stations and water depth intervals (cf. Supplement S1).

[Figure]

**S4** Figure 5**:** Laser ablation ICP-MS profiles of Mg/Ca through *G. menardii*. Single chambers (F, F-1 and F-2) were measured from the outside of the tests toward the inside (left to right). Single individuals were collected at different stations and water depth intervals (cf. Supplement S1).

[Figure]

**S4** Figure 6: Laser ablation ICP-MS profiles of Mg/Ca through *G. ungulata*. Single chambers (F, F-1 and F-2) were measured from the outside of the tests toward the inside (left to right). Single individuals were collected at different stations and water depth intervals (cf. Supplement S1).

[Figure]

**S4** Figure 7**:** Laser ablation ICP-MS profiles of Mg/Ca through *G. truncatulinoides* dextral. Single chambers (F, F-1, F-2 and F-3) were measured from the outside of the tests toward the inside (left to right). Single individuals were collected at different stations and water depth intervals (cf. Supplement S1).

[Figure]

**S4** Figure 8**:** Laser ablation ICP-MS profiles of Mg/Ca through *G. tumida*. Single chambers (F, F-1 and F-2) were measured from the outside of the tests toward the inside (left to right). Single individuals were collected at different stations and water depth intervals (cf. Supplement S1).

[Figure]

**Plate 1:** Scanning electron micrographs (SEM)

(a) *G. truncatulinoides* dextral (from station 221-8 in 150–210 m water depth)

(b) *O. universa* (from station 221-8 in 60–150 m water depth)

(c) *G. sacculifer* (from station 211-5 in 0–60 m water depth)

(d) *P. obliquiloculata* (from station 221-7 in 40–60 m water depth)

(e) *G. ungulata* (from station 211-5 in 0–60 m water depth)

(f) *G. menardii* (from station 221-7 in 0–40 m water depth)

(g) *G. tumida* (from station 219-7 in 220–400 m water depth)

Scale: 200 μm; The holes point to the spots from laser ablations in chamber F to F-3.

[Figure]

**S6** Figure 1**:** Average stable oxygen isotopes of living planktic foraminifers and fossil tests. Living foraminiferal $\delta^{18}O_{calcite}$ samples are plotted at the mean sampling depth interval. Coloured bars indicate the average weighted living depth for each species (see Jentzen et al., 2018). Black lines: $\delta^{18}O_{equilibrium}$ of the ambient seawater.

[Figure]

**S6 Figure 2:** Average Mg/Ca values (± standard deviations) of LA-ICP-MS measurements of single tests plotted at the mean sampling depth interval. Coloured bars indicate the average weighted living depth for each species (see Jentzen et al., 2018). Black lines: Temperature of the ambient seawater.

**Supplement References**

Anand, P., Elderfield, H., and Conte, M. H.: Calibration of Mg/Ca thermometry in planktonic foraminifera from a sediment trap time series, Paleoceanography, 18, 1050, doi:10.1029/2002PA000846, 2003.

Bemis, B. E., Spero, H. J., Bijma, J., and Lea, D. W.: Reevaluation of the oxygen isotopic composition of planktonic foraminifera: Experimental results and revised paleotemperature equations, Paleoceanography, 13, 150–160, doi:10.1029/98PA00070, 1998.

Bouvier-Soumagnac, Y., and Duplessy, J. C.: Carbon and oxygen isotopic composition of planktonic foraminifera from laboratory culture, plankton tows and Recent sediment: implications for the reconstruction of paleoclimatic conditions and of the global carbon cycle, The Journal of Foraminiferal Research, 15, 302–320, 1985.

Cléroux, C., Cortijo, E., Anand, P., Labeyrie, L., Bassinot, F., Caillon, N., and Duplessy, J.-C.: Mg/Ca and Sr/Ca ratios in planktonic foraminifera: Proxies for upper water column temperature reconstruction, Paleoceanography, 23, PA3214, doi:10.1029/2007PA001505, 2008.

Dekens, P. S., Lea, D. W., Pak, K., and Spero, H. J.: Core top calibration of Mg/Ca in tropical foraminifera: Refining paleotemperature estimation, Geochemistry, Geophysics, Geosystems, 3, doi:10.1029/2001GC000200, 2002.

Elderfield, H., and Ganssen, G.: Past temperature and $\delta^{18}O$ of surface ocean waters inferred from foraminiferal Mg/Ca ratios, Nature, 405, 442–445, 2000.

Erez, J., and Luz, B.: Experimental paleotemperature equation for planktonic foraminifera, Geochimica et Cosmochimica Acta, 47, 1025–1031, 1983.

Farmer, E. C., Kaplan, A., deMenocal, P. B., and Lynch-Stieglitz, J.: Corroborating ecological depth preferences of planktonic foraminifera in the tropical Atlantic with the stable oxygen isotope ratios of core top specimens, Paleoceanography 22, PA3205, doi: 10.1029/2006PA001361, 2007.

Hammer, Ø., Harper, D. A. T., and Ryan, P. D.: PAST: Paleontological statistics software package for education and data analysis, Palaeontologia Electronica 4, p. 9, 2001.

Jentzen, A., Schönfeld, J., and Schiebel, R.: Assessment of the effect of increasing temperature on the ecology and assemblage structure of modern planktic foraminifers in the Caribbean and surrounding seas, Journal of Foraminiferal Research, 48, 251–272, doi:10.2113/gsjfr.48.3.251, 2018.

Kim, S.-T., and O´Neil, J. R.: Equilibrium and nonequilibrium oxygen isotope effects in synthetic carbonates, Geochimica et Cosmochimica Acta, 61, 3461–3475, 1997.

Lea, D. W., Mashiotta, T. A., and Spero, H. J.: Controls on magnesium and strontium uptake in planktonic foraminifera determined by live culturing, Geochimica et Cosmochimica Acta, 63, 2369–2379, 1999.

McKenna, V. S., and Prell, W. L.: Calibration of *Globorotalia truncatulinoides* (R) for the reconstruction of marine temperature gradients, Paleoceanography, 19, PA2006, doi:10.1029/2000PA000604, 2004.

Mulitza, S., Boltovskoy, D., Donner, B., Meggers, H., Paul, A., and Wefer, G.: Temperature: $\delta^{18}O$ relationships of planktonic foraminifera collected from surface waters, Palaeogeography, Palaeoclimatology, Palaeoecology, 202, 143–152, 2003.

Nürnberg, D., Bijma, J., and Hemleben, C.: Assessing the reliability of magnesium in foraminiferal calcite as a proxy for water mass temperatures, Geochimica et Cosmochimica Acta, 60, 803–814, 1996.

Nürnberg, D., Müller, A., and Schneider, R. R.: Paleo-sea surface temperature calculations in the equatorial east Atlantic from Mg/Ca ratios in planktic foraminifera: A comparison to sea surface temperature estimates from $U_{37}^{K}$, oxygen isotopes, and foraminiferal transfer function, Paleoceanography, 15, 124–134, doi:10.1029/1999PA000370, 2000.

Regenberg, M., Nürnberg, D., Steph, S., Groeneveld, J., Garbe-Schönberg, D., Tiedemann, R., and Dullo, W. C.: Assessing the effect of dissolution on planktonic foraminiferal Mg/Ca ratios: Evidence from Caribbean core tops, Geochemistry, Geophysics, Geosystems 7, Q07P15, doi:10.1029/2005GC001019, 2006.

Regenberg, M., Steph, S., Nürnberg, D., Tiedemann, R., and Garbe-Schönberg, D.: Calibrating Mg/Ca ratios of multiple planktonic foraminiferal species with δ18O-calcification temperatures: Paleothermometry for the upper water column, Earth and Planetary Science Letters, 278, 324–336, 2009.

Russel, A. D., Hönisch, B., Spero, H. J., and Lea, D. W.: Effects of seawater carbonate ion concentration and temperature on shell U, Mg, and Sr in cultured planktonic foraminifera, Geochimica et Cosmochimica Acta, 68, 4347–4361, 2004.

Schönfeld, J., Bahr, A., Bannert, B., Bayer, A. S., Bayer, M., Beer, C., Blanz, T., Dullo, W. C., Flögel, S., Garlichs, T., Haley, B., Hübscher, C., Joseph, N., Kučera, M., Langenbacher, J., Nürnberg, D., Ochsenhirt, W. T., Petersen, A., Pulm, P., Titschack, J., and Troccoli, L.: Surface and Intermediate water hydrography, planktonic and benthic biota in the Caribbean Sea - Climate, Bio and Geosphere linkages (OPOKA), Cruise No. 78, Leg 1, February 22 - March 28, 2009, Colón (Panama) - Port of Spain (Trinidad and Tobago), METEOR-Berichte, 1–196, 2011.

Shackleton, N. J.: Attainment of isotopic equilibrium between ocean water and the benthonic foraminifera genus Uvigerina: Isotopic changes in the ocean during the last glacial, Colloques Internationaux du C.N.R.S. 219, 203–209, 1974.

Spero, H. J., Mielke, K. M., Kalve, E. M., Lea, D. W., and Pak, D. K.: Multispecies approach to reconstructing eastern equatorial Pacific thermocline hydrography during the past 360 kyr, Paleoceanography, 18, 1022, doi:10.1029/2002PA000814, 2003.

Steph, S., Regenberg, M., Tiedemann, R., Mulitza, S., and Nürnberg, D.: Stable isotopes of planktonic foraminifera from tropical Atlantic/Caribbean core-tops: Implications for reconstructing upper ocean stratification, Marine Micropaleontology, 71, 1–19, 2009.

---

## Author Response (AR1)

Dear Dr. Lennart de Nooijer

We kindly thank you for your effort in reviewing our manuscript. We have included all minor comments in the revised version of our manuscript.

Yours sincerely
Anna Jentzen and on behalf of all co-authors

**Minor Comments:**

page 1, line 36/37: even though it is shown for a species not studied by you, please include a reference to Kozdon et al. (2009; Chem Geol) as an example how life stage (encrustation) can influence oxygen isotopes.
We added this information P2L1-2: Additionally, encrustation of foraminiferal tests, at the end of the life cycle, results in higher $\delta^{18}O_{calcite}$ compared to non-encrusted specimens (e.g. Kozdon et al., 2009).

page 2, line 7: 'Russell'
Done.

page 2, line 34 and further: please use 'rose Bengal' instead of 'Rose Bengal'. I assume the rB was dissolved in the ethanol: was the concentration of rB in the ethanol 2 g/L? or 4 g/L (and hence 2 g/L after mixing with 50% seawater?).
We added this information. 2g rB was dissolved in 1 L ethanol and then it was added to the seawater sample.

Figure 1: could you reverse the temperature scale bar? From the caption it is not (immediately) clear what the numbers at some stations refer to. Please include e.g. by referring to Table 1.
Done.

page 4, line 21: replace 'and' by 'or'.
Done.

page 4, line 22: replace the second 'and' by 'or'.
Done.

Figures 2 and 3: salinity has no unit.
We have deleted "psu" in the text, figures and supplement.

page 5, line 20-22: this sentence appears superfluous to me: consider removing.
Done.

page 5, line 26 and elsewhere: salinity has no unit.
Done.

page 9, line 15: 'Russell'
Done.

page 9, line 31: 'calcifies'
Done.

page 11, line 23: I don't need to be mentioned in the acknowledgements.
Done.